behaviour, ecology

phenology, seasonality, predation, consumption, trade-off, climate change

**Author for correspondence:**
Roxanne S. Beltran
e-mail: roxanne@ucsc.edu

# Seasonal resource pulses and the foraging depth of a Southern Ocean top predator

Roxanne S. Beltran[1,2,3], A. Marm Kilpatrick[3], Greg A. Breed[4], Taiki Adachi[5], Akinori Takahashi[6], Yasuhiko Naito[6], Patrick W. Robinson[3], Walker O. Smith Jr[7,8], Amy L. Kirkham[9] and Jennifer M. Burns[2,10]

[1]Department of Biology and Wildlife, University of Alaska Fairbanks, 2090 Koyukuk Drive, Fairbanks, AK 99775, USA
[2]Department of Biological Sciences, University of Alaska Anchorage, 3101 Science Circle, Anchorage, AK 99508, USA
[3]Department of Ecology and Evolutionary Biology, University of California Santa Cruz, 130 McAllister Way, Santa Cruz, CA 95060, USA
[4]Institute of Arctic Biology, University of Alaska Fairbanks, P.O. Box 757000, Fairbanks, AK 99775, USA
[5]Department of Biological Sciences, University of Tokyo, 2-11-16 Yayoi, Bunkyō, Tokyo 113-0032, Japan
[6]National Institute of Polar Research, 10-3 Midori-cho, Tachikawa, Tokyo 190-8518, Japan
[7]Virginia Institute of Marine Science, College of William and Mary, 1375 Greate Rd, Gloucester Point, VA 23062, USA
[8]Institute of Oceanography, Shanghai Jiao Tong University, 1954 Huashan Road, Shanghai, 200240, People's Republic of China
[9]College of Fisheries and Ocean Sciences, University of Alaska Fairbanks, 17101 Point Lena Loop Road, Juneau, AK 99801, USA
[10]Department of Biological Sciences, Texas Tech University, Box 43131 Lubbock, TX 79409, USA

RSB, 0000-0002-8520-1105; AMK, 0000-0002-3612-5775

Seasonal resource pulses can have enormous impacts on species interactions. In marine ecosystems, air-breathing predators often drive their prey to deeper waters. However, it is unclear how ephemeral resource pulses such as near-surface phytoplankton blooms alter the vertical trade-off between predation avoidance and resource availability in consumers, and how these changes cascade to the diving behaviour of top predators. We integrated data on Weddell seal diving behaviour, diet stable isotopes, feeding success and mass gain to examine shifts in vertical foraging throughout ice break-out and the resulting phytoplankton bloom each year. We also tested hypotheses about the likely location of phytoplankton bloom origination (advected or produced *in situ* where seals foraged) based on sea ice break-out phenology and advection rates from several locations within 150 km of the seal colony. In early summer, seals foraged at deeper depths resulting in lower feeding rates and mass gain. As sea ice extent decreased throughout the summer, seals foraged at shallower depths and benefited from more efficient energy intake. Changes in diving depth were not due to seasonal shifts in seal diets or horizontal space use and instead may reflect a change in the vertical distribution of prey. Correspondence between the timing of seal shallowing and the resource pulse was variable from year to year and could not be readily explained by our existing understanding of the ocean and ice dynamics. Phytoplankton advection occurred faster than ice break-out, and seal dive shallowing occurred substantially earlier than local break-out. While there remains much to be learned about the marine ecosystem, it appears that an increase in prey abundance and accessibility via shallower distributions during the resource pulse could synchronize life-history phenology across trophic levels in this high-latitude ecosystem.

# 1. Background

Across the globe, the loss of predators [1,2] and shifting resource availability due to climate change and habitat alterations [3,4] have underscored the importance of resource pulses in structuring spatial use by various trophic levels [5,6]. In some cases, predators alter the distribution of their prey directly via consumption [7–10] or indirectly through cascading species interactions [11]. In other cases, environmental factors such as nutrient availability regulate the distribution of primary producers and their consumers [12]. A key question is how resource pulses impact the fine-scale behaviour and energetics of top predators [13].

This question is especially relevant in polar regions where strong physical forcing of retreating sea ice triggers a short-term phytoplankton bloom at the ocean's surface each year [14–17]. The approximately 20-fold increase in phytoplankton biomass concentrates zooplankton and fishes in the marginal ice zone [18–22]. This resource pulse facilitates reproduction [23] and survival [24] of individuals across species and trophic levels [25]. However, environmental conditions are changing rapidly and increasingly differ from those in which life-history strategies evolved [26]. A major concern is whether climate change will alter species interactions to create temporal or spatial mismatches that compromise individual fitness [27,28]. It is therefore important to understand how ecological dynamics vary across time and three-dimensional space.

A fundamental characteristic of ocean ecosystems is that they are vertically stratified, with depth gradients of temperature, light, nutrients and oxygen that constrain biological processes [29]. Primary producers such as phytoplankton require sunlight for photosynthesis, which constrains their distributions to approximately the upper 50 m of the ocean. Additionally, air-breathing predators such as seals, whales, and seabirds must return to their oxygen supply at the surface after foraging underwater. Behavioural theory suggests that intermediate trophic levels such as zooplankton and fishes will maximize fitness by balancing ecological trade-offs between resource acquisition and predation risk, which both vary with depth [30,31]. Empirical evidence suggests that on a daily timescale, most zooplankton and fishes balance this trade-off by consuming phytoplankton near the surface during the night and descending to depth to avoid visual predators during the day. These diel vertical migrations [32] are often tracked by air-breathing vertebrates [33–35]; however, it is unclear whether ephemeral resource pulses cause analogous cascading migrations on seasonal timescales [36].

Our aim was to understand how sea ice break-out and resource pulses influence top predators' use of vertical space at a study site in the southern Ross Sea, which is in the most productive sector of the Southern Ocean in Antarctica [37,38]. We integrated satellite-observed ice dynamics with the behaviour, diet and diving efficiency of a top predator (the Weddell seal, *Leptonychotes weddellii*) to infer temporal variation in the vertical distribution of lower trophic levels throughout the austral summer. We hypothesized that during spring, when resource stratification was weak, intermediate trophic levels (zooplankton and fish) would have deeper distributions as compared to summer, when sea ice break-out triggers a strong resource pulse of phytoplankton at the ocean surface. This resource pulse could increase the efficiency of energy transfer to air-breathing top predators, which must return to the surface for oxygen after foraging underwater.

# 2. Methods

## (a) Study design

We used archival biologgers to record the diving and foraging behaviour of 59 adult female Weddell seals in Erebus Bay, Antarctica in austral summers 2013–2016 (hereafter, AS13–AS16). All study animals were between the ages of 10 and 20 years and had given birth at least once prior to inclusion in this study [39]. Each individual was chemically immobilized during the November and December lactation period as described in Shero, Pearson [40] and instrumented with a time-depth recorder (hereafter TDR, manufactured by LOTEK, model LAT1800, 6 s sampling interval, 0.5 m depth resolution) and a VHF tag for relocation (manufactured by SIRTRACK) on flipper tags [41]. In addition, in AS16 (Nov 2016–Feb 2017), we used Loctite epoxy to affix Little Leonardo acceleration loggers (measuring two-axis raw acceleration at 20 Hz) to the lower jaws of four seals. We recaptured seals and removed the TDRs 57 ± 13 days later and accelerometers 2–4 days later. Seals were weighed at tag deployment and recovery.

## (b) Ice break-out and phytoplankton bloom dates

Direct measurements of phytoplankton bloom timing were not available throughout our study site and the timing of ice break-out and the resulting phytoplankton bloom are generally not well understood [38]. Because advective (i.e. allochthonous) inputs are thought to contribute significantly more to water column productivity than local *in situ* (i.e. autochthonous) production [42], we considered a suite of possible phytoplankton sources (five areas around Ross Island) and current velocities (three rates) as advective drivers of the seal diving patterns. To do this, we obtained satellite-derived daily sea ice concentration (% cover; US National Snow and Ice Data Center; NASA Bootstrap Sea Ice Concentrations from Nimbus-7 SMMR and DMSP SSM/I-SSMIS, Version 3; spatial resolution $25 \times 25$ km) and defined the date of sea ice break-out as the first occurrence of a 7-day running mean ice concentration of less than 50% at each gridded data location [43] (electronic supplementary material, figure S1) which we labelled 'Northeast' (centred at 77.24° S 169.10° E), 'North' (77.20° S 168.09° E), 'Northwest' (77.10° S 166.09° E), 'West' (77.32° S 165.84° E) and 'Southwest' (77.54° S 165.58° E). Next, we estimated the number of days it would take for phytoplankton to advect into the Erebus Bay study region (77.6° S 167.0° E) [42] from each data location based on three previously measured ocean current velocities: 12 km day$^{-1}$ [44], 10.3 km day$^{-1}$ [45] and 6.5 km day$^{-1}$ [46]. Hereafter in the text and figures, we use the term 'resource pulse' to describe the approximately 35-day phytoplankton bloom that begins in Erebus Bay on the date of advective arrival from around Ross Island (electronic supplementary material, figure S2, [38]).

## (c) Statistical analysis

For each dive, we calculated the maximum depth, total duration, bottom phase duration (defined as 80% of maximum dive depth) and number of bottom wiggles (inflection points in the depth profile of the bottom phase) using the IKNOS toolbox in MATLAB (Y Tremblay 2005, unpublished) [47,48]. We identified prey capture attempts within the raw acceleration data using a surge acceleration threshold of 0.3 g [49] (electronic supplementary material, figure S3). We classified each dive as either benthic or pelagic as previously described [50], using parameter thresholds specified in the electronic supplementary material. We used a linear mixed-effects model in R (package 'lme4' v. 1.1–14) to determine whether benthic dive depths varied predictably across the summer. Benthic dives (1% of all dives) were excluded from all remaining analyses because we were interested

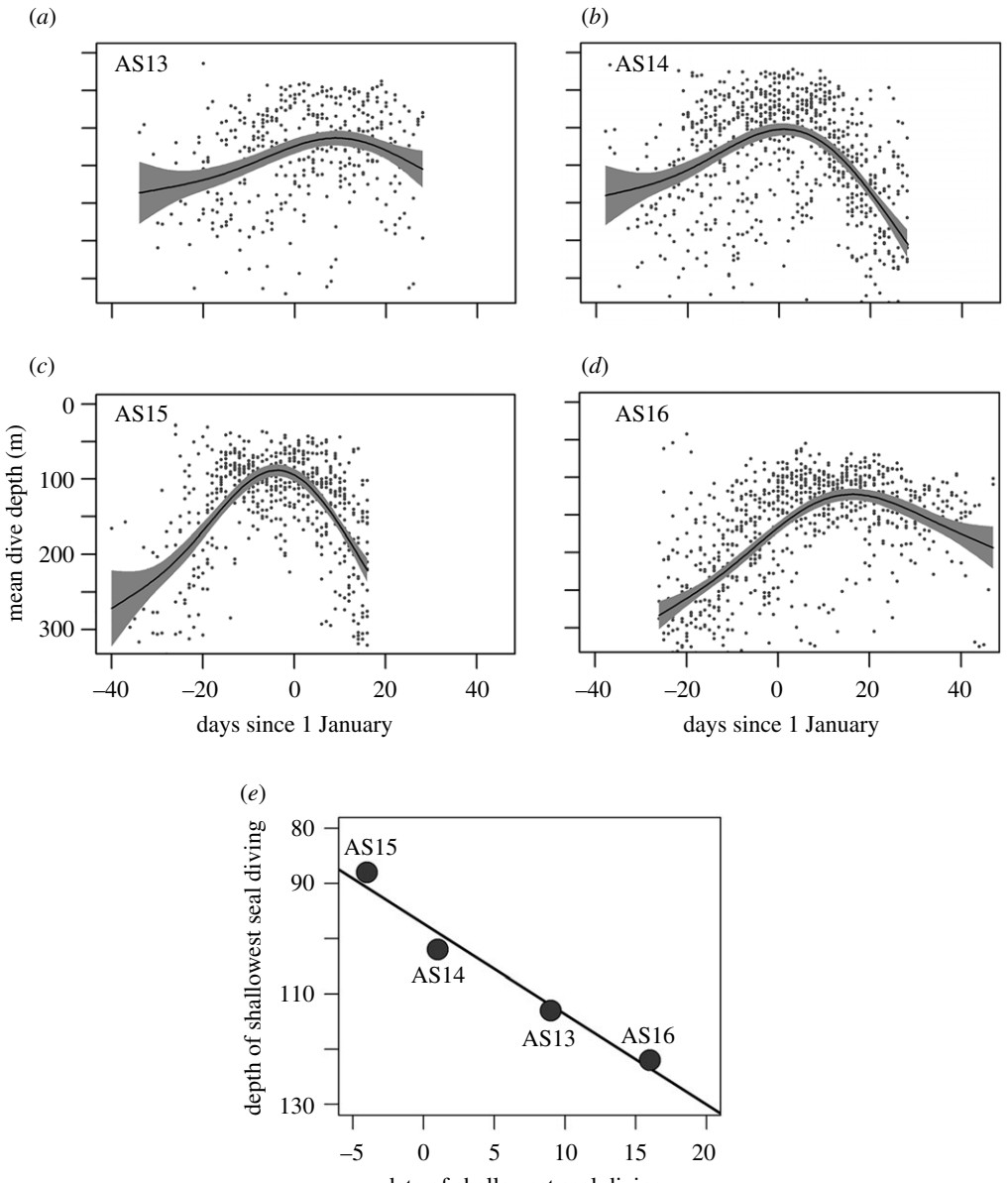

**Figure 1.** Patterns of seal dive depth and date across four years. (*a,b,c,d*) Generalized additive mixed models (lines) and 95% CI fits (polygons) for mean dive depth in metres (each point represents mean dive depth for one seal on 1 day) across the austral summer (AS) in 2013, 2014, 2015 and 2016. Data represent 59 seals. (*e*) Correlation between Julian day of shallowest diving (days since 1 January) and depth of shallowest diving in metres. Note the reversed *y*-axis with zero at the top of the plot representing the ocean surface.

in quantifying seasonal changes in mid-water dives. We then characterized seasonal changes in diving depth by calculating the mean across all seals of the maximum dive depth on each day for each seal ($n = 2941$ seal-days; figure 1 and electronic supplementary material, figure S4). Finally, we evaluated the inter-annual differences in benthic diving depth by fitting a fourth order generalized additive mixed model (GAMM) to all dives with the individual seal as a random effect. We used a GAMM with the individual as a random effect to model foraging depth as a function of calendar date, ice break-out date and phytoplankton advection arrival date. Akaike information criteria (AIC) model selection methods were used to compare the relative strength of models.

## (d) Seal diet

In austral summers 2013 and 2014, we plucked one whisker from each of nine seals during tag attachment and plucked the regrown whisker during tag removal. To infer seal diet across the summer period, we analysed carbon $\delta^{13}$C and nitrogen $\delta^{15}$N stable isotope values in sequential segments of whiskers grown during the biologger deployment period. We

incorporated average whisker $\delta^{13}$C and $\delta^{15}$N values for each seal with stable isotope values of five published prey groups (electronic supplementary material, table S1, [51]) into a stable isotope mixing model using the *R* package SIAR [52]. We then examined a time-series of $\delta^{13}$C and $\delta^{15}$N values from deployment to recovery by assigning timestamps to each whisker segment as described and validated in Beltran, Sadou [53]. Using linear mixed-effects models, we characterized the relationship between date (days since 1 January) and stable isotope ratios ($\delta^{13}$C and $\delta^{15}$N) in whiskers with the individual as a random effect. Full methodological details are provided in the electronic supplementary material.

## 3. Results

## (a) Vertical space use of top predators

We analysed 138 506 dives from 59 Weddell seals to characterize the diving depth and foraging effort of each seal across the austral summer over 4 years (AS13–AS16; figure 1). Mean dive depth in early summer was 233 m with only 39%

of dives being less than 200 m, suggesting that prey species were found at depth. As sea ice extent decreased throughout the summer, mean seal dive depth shallowed to 110 m and 83% of dives occurred within 200 m of the surface. The seal shallowing period, when mean diving depth across seals did not exceed 140 m, lasted approximately 24 days. Subsequently, dive depths deepened back to 230 m (with 41% being above 200 m), indicating prey had shifted back to depth (figure 1). There was clear inter-annual variation in seal diving behaviour, with shallowing occurring earliest in AS15 and latest in AS16 (figure 1). Across years, seal diving was shallower when the date of seal shallowest seal diving was earlier (figure 1; $Depth_{shallowestdives} = 1.64$ ($\pm$ SE $= 0.18$) $\times Date_{shallowestdives} + 97.25$, $n = 4$ years, $R^2 = 0.95$, one-tailed $p = 0.015$).

## (b) Spatial distribution of foraging

The depth of benthic dives suggested that seals foraged in the same area throughout the austral summer, which aligns with findings from a prior mark–recapture study [39]. Benthic dive depths were stable over time (electronic supplementary material, figure S6) and matched the bathymetry of the nearby Erebus Bay area [54].

## (c) Consistency in seal diets

Changes in diving depth were not due to seasonal shifts in seal diets. Seal whisker $\delta^{15}N$ and $\delta^{13}C$ isotope values showed no consistent trends over time, suggesting that seals did not shift their diet across summer (figure 2). A stable isotope mixing model of seal whiskers indicated that seals consistently consumed silverfish *Pleurogramma antarcticum* and *Trematomus newnesi* (diet proportion 72% (95% CI: 34%–86%) with smaller amounts of *Pagothenia borchgrevinki* and other *Trematomus* spp. (diet proportion 20%, 95% CI: 6% to 39%) (figure 2 and electronic supplementary material, table S1).

## (d) Diving efficiency of top predators

Shallower diving was associated with increased seal feeding rates, diving efficiency and mass gain. Two proxies for prey capture were correlated with each other: jaw motion events (feeding attempts associated with quick mouth opening) and depth wiggles (vertical excursions in a dive's bottom phase, where feeding usually occurs) [49] (electronic supplementary material, figure S5A). In turn, wiggles were a strong predictor of mass gain per hour diving (electronic supplementary material, figure S5B) and were markedly more frequent during the shallow period ($WiggleRate = -0.0043$ ($\pm$ SE $= 0.0002$) $\times MeanDiveDepth + 3.2589$, $R^2 = 0.83$, $p < 0.0001$, $n = 76$ days). Feeding rates increased by 16% from the first 10 deep days to the shallow period and then decreased by 11% in the last 10 deep days (figure 3). In addition, the proportion of time spent in the bottom phase of each dive where most feeding occurs (hereafter, diving efficiency) increased 66% from early to mid-summer and then decreased 19% thereafter (figure 3). The cumulative effects of increased feeding rates in the bottom phase and spending proportionally more time on each dive in the bottom phase resulted in an 82% increase in diving efficiency during the mid-summer shallow period compared to the deep period in early summer.

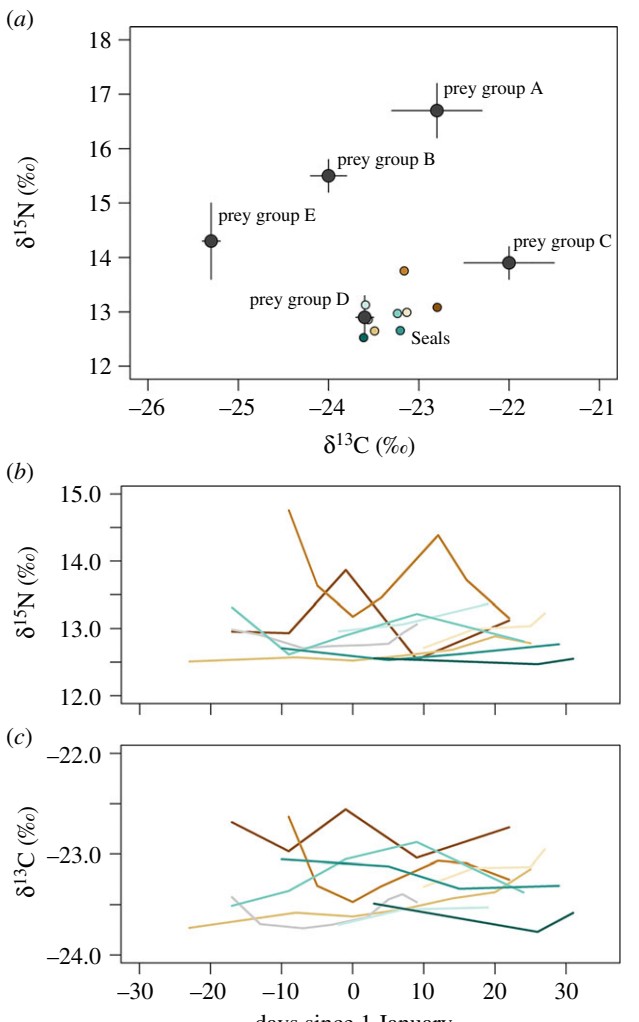

**Figure 2.** Stable isotope analysis of diet suggested no seasonal diet switch. (*a*) Stable isotope values (mean $\pm$ 1 s.d.) of Weddell seal whisker samples (coloured circles, our study) and published prey groups (grey circles, see electronic supplementary material, table S1) adjusted using trophic enrichment factors. Temporal variation in seal whisker $\delta^{15}N$ (*b*) and $\delta^{13}C$ (*c*) values across summer. Seal values were measured in regrown whiskers from nine individuals that gained mass over summer ($n = 47$ whisker segments, mean $\pm$ s.d.; $\delta^{15}N$ 13.0 $\pm$ 0.5‰ and $\delta^{13}C$ $-23.2 \pm 0.3$‰). Each line in (*b*) and (*c*) represents an individual seal.

## (e) Links with sea ice break-out and resource pulses

As expected, sea ice break-out progressed anticlockwise around Ross Island from the Northeast to the Southwest over approximately 40 days, although ice break-out progression showed marked inter-annual variation (figure 4 and electronic supplementary material, figure S1). Advection from the five data locations to the study location took 5 to 26 days based on the slow current velocity [46], 3 to 17 days based on the intermediate current velocity [45], and 3 to 14 days based on the fast current velocity [44] (figure 4). We note that these advection durations represent a rough approximation given that inter-annual variations in current speed are expected from large-scale atmospheric and oceanic processes as well as depth, tides and ice dynamics [55].

We did not find consistent support for the hypothesis that seal shallowing is triggered by ice break-out or arrival of advected phytoplankton from a particular source. The period of shallowest diving in each year estimated by the GAMMs was tightly correlated with the temporal arrival of

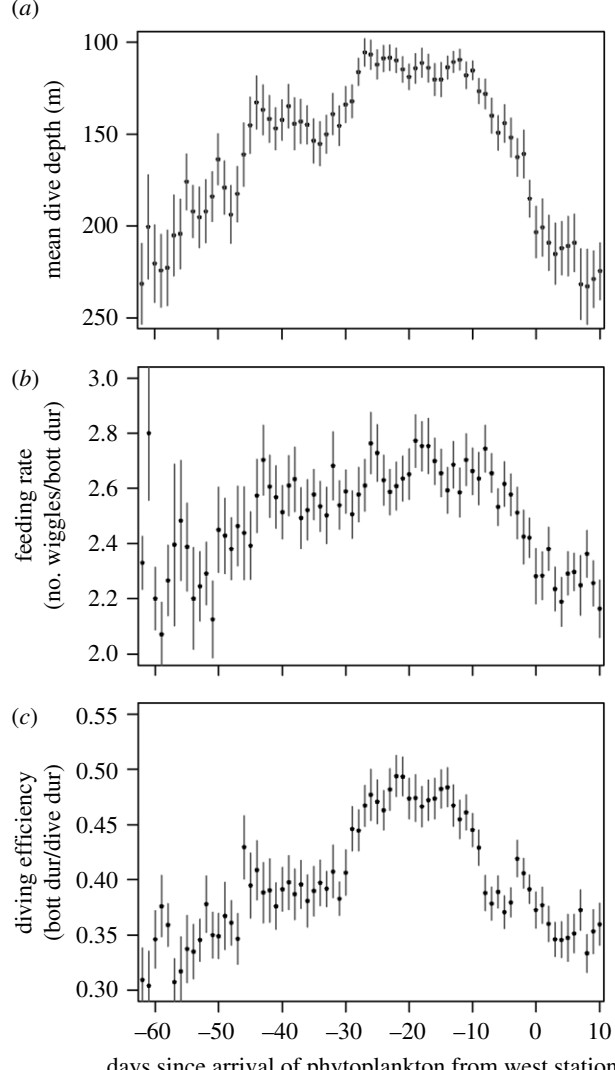

**Figure 3.** Temporal variation in dive depths, feeding rates and diving efficiency across summer relative to the arrival date of the resource pulse from the West. Mean and standard error of dive depth (*a*), feeding rate, measured in wiggles per minute bottom time (*b*), and diving efficiency, measured in proportion of the dive duration comprised the bottom phase, among all seals for each day (*c*). Data represent 59 seals observed between 2013 and 2016.

the resource pulse for some seasons (e.g. AS13) but not others (e.g. AS15) and from some data locations (e.g. West) but not others (e.g. Northeast) (figure 4, bottom). In fact, the seal shallowing occurred before ice break-out at three data locations and before calculated phytoplankton arrival from four data locations (figure 5b). One possible explanation is that the Cape Royds current around Ross Island (figure 4a) may be notably faster than the currents near the seal colony that we used in our estimates [56]. Interestingly, the rate of advection exceeded that of ice break-out, which could explain why local break-out occurs substantially later than seal dive shallowing (figure 5). The inter-annual variability in seal dive shallowing was most explained (26% of variance) by the two data locations closest to the Erebus Bay study area, West and Southwest (figure 5b). Additionally, seal foraging depth was better explained by the date of advected phytoplankton arrival than calendar date (GAMMs; ΔAIC = 490) for all but one data location, Northeast (figure 5b), that showed extremely limited inter-annual variability in ice break-out dates (electronic supplementary

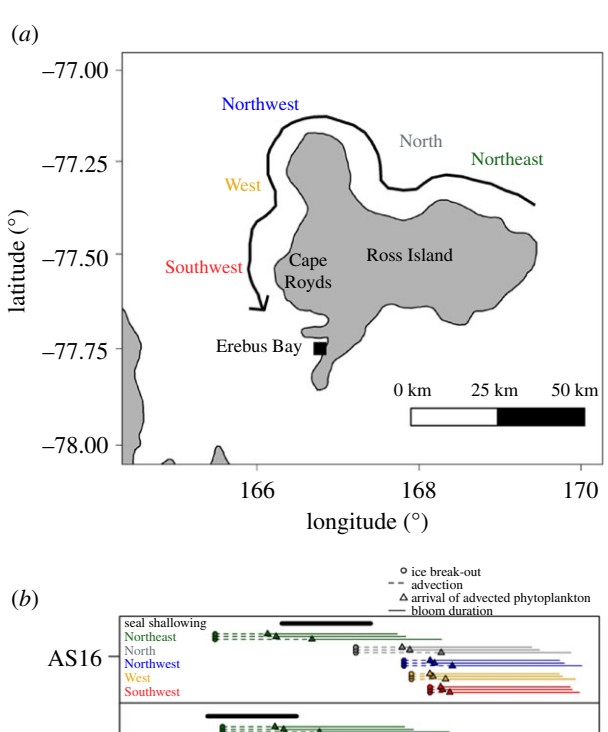

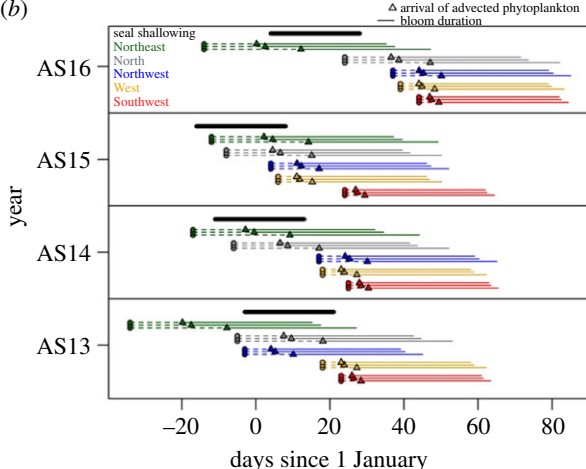

**Figure 4.** Temporal and spatial patterns of ice break-out, phytoplankton advection and bloom duration for each of five locations using three published current velocities in the four study years. (*a*) Sea ice break-out (7-day running mean of less than 50% ice cover) and phytoplankton advection follow the coastline from Northeast to Southwest, reaching the Erebus Bay study site (black square) last. (*b*) Seal shallow diving (thick black line) overlayed on ice break-out (circles), phytoplankton advection (dashed lines), arrival of phytoplankton advection (triangles) and approximate bloom duration (solid lines) for each ice location (colours) and year (panels). Each location/year combination has three lines representing various estimates of ocean current velocities, from top to bottom: 12 km day$^{-1}$ [44], 10.3 km day$^{-1}$ [45], and 6.5 km day$^{-1}$ [46].

material, figure S1). However, the seal shallowing preceded phytoplankton advection by up to 20 days across the four study years, suggesting that phytoplankton advection arrives earlier than estimated from current velocities and ice break-out dates.

## 4. Discussion

We show that in Antarctic summer, ice break-out and the resulting resource pulse are associated with increased energy transfer to top predators. Foraging depth of Weddell seals halved and remained shallow for three weeks in mid-summer, suggesting that zooplankton and fishes may have shallowed [57], despite the predation risk from air-breathing predators (figure 6). Diving efficiency of seals (the proportion of time spent in the bottom phase of each dive) nearly

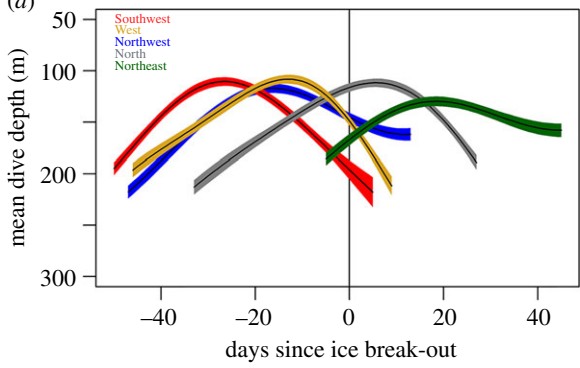

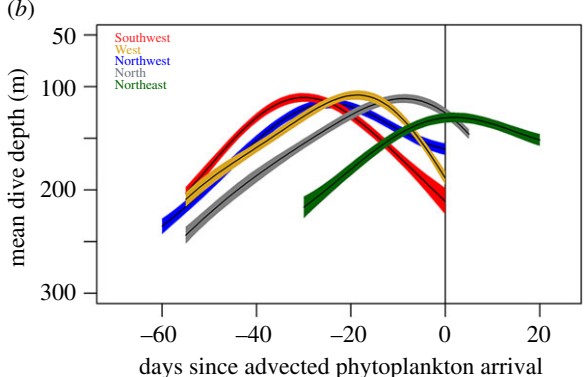

| explanatory variable | AIC | ΔAIC | $R^2$ |
|---|---|---|---|
| West | 29254 | - | 0.264 |
| Southwest | 29256 | 2 | 0.263 |
| North | 29267 | 13 | 0.260 |
| Northwest | 29473 | 219 | 0.199 |
| Julian Day | 29744 | 490 | 0.111 |
| Northeast | 29830 | 576 | 0.080 |

**Figure 5.** Across all years, temporal patterns of seal diving depth in relation to ice break-out and phytoplankton arrival from five locations (colours). Generalized additive mixed models (lines) and 95% CI fits (polygons) for mean seal dive depth across all study years in relation to (a) the date of ice break-out at each of five ice locations and (b) the approximate phytoplankton arrival to the local Erebus Bay study site from each of five ice locations. Advection duration was calculated using the intermediate current speed of 10.3 km day$^{-1}$ from Littlepage [45]. Bottom table, AIC and $R^2$ values for the GAMM fits of mean diving depth as a function of temporal patterns at each ice location.

doubled during this period because shallower dives require less descending and ascending transit time [58]. At the end of summer, seals resumed foraging at deeper depths (figure 6). The deepening of dives may coincide with iron limitation [59] and phytoplankton biomass depletion (electronic supplementary material, figure S2), when mesopelagic fishes [22,60] and krill [19] are thought to inhabit deeper depths. Across the summer there was little change in seal diet composition or horizontal space use, indicating that seal diving patterns reflect shifts in prey vertical distributions. Together, these data suggest that sea ice break-out and the associated resource pulse triggers cascading vertical distribution changes of three trophic levels: zooplankton, fishes and seals.

We found substantial variation in timing between seal shallowing and the resource pulse that cannot be explained by our existing understanding of the complex marine ecosystem (figure 4). Several critical questions about ecosystem-

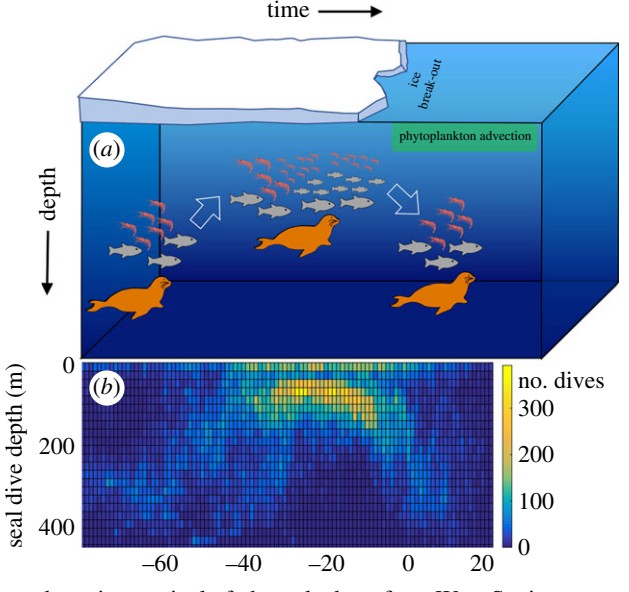

**Figure 6.** The annual ice break-out and phytoplankton bloom shifts three-dimensional space use across trophic levels. (a) Conceptual figure of changes in seal foraging depths, fish and zooplankton (such as krill) relative to sea ice break-out and the associated resource pulse (green box) across the summer season. (b) Diving depths of Weddell seals across all years.

level processes remain. Is 50% ice cover the threshold at which ice break-out occurs and phytoplankton production commences *in situ*? How does chlorophyll concentration change when ice cover drops below 50% then re-freezes (electronic supplementary material, figure S1)? Do advection rates around Ross Island exceed those directly measured in Erebus Bay [56]? What proportion of phytoplankton is consumed in the Cape Royds current before the bloom reaches Erebus Bay? What are the relative contributions of advected phytoplankton versus *in situ* phytoplankton production, and from which advective source(s)? How long does the resource pulse on the west side of Ross Island last? Is the spatial resolution of satellite-derived ice data (25 × 25 km) sufficient to describe this highly dynamic ecosystem? Alternatively, the mismatch could be due to ecological processes, such as seal prey avoiding peak resource aggregations to optimize risk-reward trade-offs, which in turn might contribute to the observed mismatch between seal shallowing and the arrival of advected phytoplankton [61].

Regardless of the phytoplankton source and precise arrival timing, summer is clearly a season of opportunity for these top predators with important implications for energetics and behaviour that cascades up the food chain. In the future, these complex dynamics could be studied *in situ* by instrumenting seals with satellite-linked conductivity–temperature–depth–fluorescence tags [62].

Previous studies have shown that zooplankton and silver-fish exhibit seasonal differences in depth distributions [57,63] and that downward movement (i.e. vertical 'retreat') of zooplankton coincides with the end of the phytoplankton bloom in other Southern Ocean regions [64–66]. Our study links these findings together and demonstrates their influence on the behaviour and energetics of top predators. Specifically, the simultaneous shallowing of lower trophic levels during phytoplankton blooms allows a suite of air-breathing predators to feed on prey species that are often inaccessible. During

phytoplankton blooms, surface-feeding Antarctic seabirds, including snow petrels, *Pagodroma nivea,* and Antarctic petrels, *Thalassoica antarctica,* consume mesopelagic fishes [67,68] that normally occur at depths greater than their diving limits [67]. Similarly, Adélie penguins, *Pygoscelis adeliae,* which rarely dive below 70 m [69], increase the proportion of silverfish *P. antarcticum* in their diet eightfold during the phytoplankton bloom [70]. During the phytoplankton bloom in the Arctic, little auks, *Alle alle,* also consume zooplankton which normally exist deeper than their diving abilities [71].

The short-lived absence of resource limitation and shallowing of intermediate trophic level species also appears to synchronize critical life-history events across multiple trophic levels. Krill, copepods and amphipods all spawn synchronously in surface waters to exploit the brief resource pulse [19,72–74] despite the risk of predation from air-breathing predators. In turn, near-surface zooplankton are consumed by larval silverfishes which hatch at the same time [18,63,75]. These rich aggregations of lower trophic level organisms in shallow waters also appear to drive predator reproductive phenology. In mid-summer, weaned seal pups and fledged penguin chicks with limited diving capacities can feed on the shallower silverfish, which reside at deeper depths during other seasons [63,76–78].

## 5. Conclusion

In summary, ice break-out and resource pulses in the highly seasonal Southern Ocean appear to shift vertical prey distributions from deeper to shallower waters. Vertical redistributions of food chains are analogous to the horizontal migrations documented in other ecosystems [79,80] and appear to be an important driver of reproductive phenology in many air-breathing vertebrate predators including penguins, seals and seabirds. The cascading impacts of three-dimensional resource pulses on the physiology, behaviour, ecology and evolution of marine communities creates a tightly coupled ecosystem that may be at risk of phenological disruptions. Future research should seek to understand whether the photoperiod cues used by top predators to time their breeding may fail to predict resource pulse phenology under global change [81].

**Ethics.** Research activities were approved by National Marine Fisheries Service Marine Mammal permit no. 17411, University of Alaska IACUC protocols no. 419971 and no. 854089 and the Antarctic Conservation Act permit no. 2014-003.

**Data accessibility.** Data are available at the online United States Antarctic Program Data Center (https://www.usap-dc.org/view/dataset/601338).

**Authors' contributions.** R.S.B., W.O.S. and J.M.B. conceived the manuscript and study design. R.S.B., A.L.K. and J.M.B. collected the data. R.S.B., G.A.B., A.M.K., T.A., A.T., Y.N., P.W.R. and J.M.B. analysed the data. R.S.B., A.M.K., G.A.B. and J.M.B. wrote the manuscript. All authors reviewed and edited the manuscript.

**Competing interests.** We declare we have no competing interests.

**Funding.** This work was supported by National Science Foundation (NSF) grant ANT-1246463 to J.M.B. and J.W.T.; National Geographic CRE grant no. 9802-15 to J.M.B. and R.S.B.; NSF graduate research fellowship no. 2015174455 and NIPR's International Internship Program for Polar Science to R.S.B.; an INBRE graduate research fellowship to A.L.K.; and NSF grant ANT-1443258 to W.O.S. Research reported in this publication was supported by an Institutional Development Award (IDeA) from the National Institute of General Medical Sciences of the National Institutes of Health under award number P20GM103395. The content is solely the responsibility of the authors and does not necessarily represent the official views of the National Institutes of Health. This material is based upon work supported by the NSF Graduate Research Fellowship Program (R.S.B.) and while serving at the NSF (J.M.B.). Any opinion, findings, and conclusions or recommendations expressed in this material are those of the authors and do not necessarily reflect the views of the NSF.

**Acknowledgements.** This paper is dedicated to Dr Umihiko Hoshijima, whose below-the-ice perspectives were invaluable to this research. We are thankful for B-292 team members Dr Michelle Shero and Skyla Walcott as well as Clara Woolner, Dr Casey Youngflesh, Dr Takeshi Tamura, Dr John Francis and researchers from Japan's National Institute of Polar Research (NIPR) and Little Leonardo. This project was made possible by logistical support from the NSF United States Antarctic Program, Lockheed Martin Antarctic Support Contract, and support staff in Christchurch, New Zealand and McMurdo Station.

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
