## [Peer Review File · Proceedings of the Royal Society B: Biological Sciences]

Review History

RSPB-2020-0377.R0 (Original submission)

Review form: Reviewer 1

Recommendation

Accept with minor revision (please list in comments)

Scientific importance: Is the manuscript an original and important contribution to its field?

Excellent

General interest: Is the paper of sufficient general interest?

Excellent

Quality of the paper: Is the overall quality of the paper suitable?

Excellent

Is the length of the paper justified?

Yes

Should the paper be seen by a specialist statistical reviewer?

No

Do you have any concerns about statistical analyses in this paper? If so, please specify them explicitly in your report.

No

It is a condition of publication that authors make their supporting data, code and materials available - either as supplementary material or hosted in an external repository. Please rate, if applicable, the supporting data on the following criteria.

Is it accessible?

Yes

Is it clear?

Yes

Is it adequate?

Yes

Do you have any ethical concerns with this paper?

No

Comments to the Author

This is a well-written and nicely documented paper. Well done. I've no major concerns and my only a handful of minor comments, so let's get to them.

1) It would be helpful to ensure all figures, including supplemental figures, are cited and cited in the order they are discussed in the text. For example - Fig 5a is referred to indirectly only after 5b and 5c are introduced; Figure S1 is not referenced in the text; Fig 3A is not referenced in the text; Figure S3 is referenced before any other.

2) Line 111-112 Are the dates of pupping and molt critical for this analysis? might the end of this sentence be deleted?

3) To help with figure numbering issues noted above and reference Fig 3A (indirectly, at least), I suggest you refer to Figure 3 at the end of the sentence on line 169.

4) Line 190-192. Fair point here, but I feel like a more convincing argument could be made if you can link to other ancillary evidence from prior tracking studies in the Ross Sea or elsewhere. Do those studies support your assumption that the seals remain reasonably stationary during the period of your tracking.

Line 201 - it may be worth clarifying that foraging efficiency increases because more time can be spent foraging relative to ascent and descent on shallow dives. Presumably total dive time or average dive time did not change much during across the tracking period?

Review form: Reviewer 2

Recommendation

Major revision is needed (please make suggestions in comments)

Scientific importance: Is the manuscript an original and important contribution to its field?
Acceptable

General interest: Is the paper of sufficient general interest?
Acceptable

Quality of the paper: Is the overall quality of the paper suitable?
Marginal

Is the length of the paper justified?
Yes

Should the paper be seen by a specialist statistical reviewer?
No

Do you have any concerns about statistical analyses in this paper? If so, please specify them explicitly in your report.
No

It is a condition of publication that authors make their supporting data, code and materials available - either as supplementary material or hosted in an external repository. Please rate, if applicable, the supporting data on the following criteria.

Is it accessible?
Yes

Is it clear?
Yes

Is it adequate?
No

Do you have any ethical concerns with this paper?
No

Comments to the Author

This MS investigates inter-annual differences in the progression of the dive foraging pattern of Weddell Seals and relates these to the timing of ice breakout as a proxy for phytoplankton bloom timing. It also shows, via stable isotope analysis, that the diet of these seals remains relatively constant despite changes in dive depth. This is presented as evidence of a seasonal vertical restructuring of the foodweb in response to the bottom-up effect of the "resource pulse" delivered by the phytoplankton bloom.

I found the paper mainly well written with a clear narrative. I have no concerns about the methodology. It is a perfectly good study of the relationship between diving behaviour and ice-dynamics, which is enhanced by stable isotope data showing that diet remains relatively constant. This supports the interpretation that predator dive distribution tracks prey vertical distribution. This in turn raises some interesting questions about what drives these changes. Although the authors provide convincing evidence that these changes are associated with the timing of ice break-out, their proposed mechanism does not fit the evidence. I also have some concerns about the ambiguity of language around key information about spatial scale and what constitutes a "resource pulse".

The authors have used their observations of predator diving behaviour to infer a sequence of events affecting a whole food chain. I feel that the inference stretches the data beyond its

capabilities.

The paper would be stronger if it included data to support these inferences, particularly about the phytoplankton bloom. I acknowledge that the MS provides some information (Fig S2) about the relative timing of sea ice and phytoplankton, but this is at a different location and in a different year from the study. Perhaps more clarity about the relationship between these data and the inferences the authors are making would help.

Unfortunately though, the main conclusions that the “extreme near-surface resource pulse ... resulted in prey shallowing” [40] and “sea ice break-out appears to trigger cascading vertical migrations... shifting control... to bottom up” [195] [also 206], seem to contradict the evidence of Fig 3 which shows that shallow diving (which is presented as an indicator of prey vertical distribution) begins before ice break-out. This shallow diving is described as occurring “for three weeks during the phytoplankton bloom” [198]. Yet “the ice break-out date was considered the approximate start date of in situ primary production” [135]. If anything, break-out seems to mark the beginning of a return to deeper diving.

The ice break-out information is for a location 40km distant from the study site, but there is no discussion of how representative this is of the study site. The data (Fig S2) suggest that peak production follows ice break-out and chlorophyll is lower 5 days before ice break-out than during the post-bloom period two months later (i.e. there is no evidence of a phytoplankton bloom that occurs during the period of shallow diving before ice break-out).

Fig 1 shows that minimum dive depth occurs before 1st January and is followed by a rapid return to deeper diving, whereas Fig S6 shows that ice break-out was always after 1st January. The claim that “At the end of the summer, when... phytoplankton biomass was depleted... seals resumed foraging at (deeper) depths” is not supported by this evidence. It looks like seals begin shallow foraging before midsummer and return rapidly to deep diving just as the putative bloom develops.

The authors do not explicitly acknowledge this mismatch in timing between observations and the expectation based on their assumptions. The authors allude to the possibility of earlier phytoplankton availability ([135-137], Fig S2 caption) but this is not sufficient to resolve the inconsistency. Also the language around primary production/phytoplankton bloom and timing is ambiguous. Are “in situ primary production” and “phytoplankton bloom” synonymous? How is ice break-out an “estimate of primary production” [136], and what does “conservative” mean in this context? Do you mean it is an estimate of the start of the bloom, but is likely to be late? The authors need to be clear and consistent in describing events that are central to their argument.

Also, the following section of the Supplement lacks any information which allows the reader to relate the sequence of events to the timing metrics (date and ice break-out date) used in the MS. Rather, it suggests that ocean current velocity is the primary determinant of in situ phytoplankton production timing:

“Each summer as sea ice degrades, three sequential processes occur. Initially, sea ice microalgae are released into the water column and can photosynthesize at very low irradiance levels; however, these microalgae contribute little to standing chlorophyll concentration. Next, phytoplankton are advected into the Erebus Bay study area (77.6°S 167.0°E) from the Ross Sea polynya (centered approximately at 77.0°S 175.0°E). About two weeks later (based on ocean current velocity), in situ phytoplankton production markedly increases as nutrients are released and ice no longer shades the photic zone (3).”

So, it seems that the data do not support sea ice break-out as the trigger for deep diving (in contrast to the statements on lines 195 and 206), or an “extreme resource pulse” in the form of a local phytoplankton bloom as the mechanism (in contrast to the central theme of the MS). Rather

the data suggest that both of these events mark the end of shallow diving.

The paragraph quoted above suggests that there could be some elevation of phytoplankton at the study site before ice break-out, as a result of algal release elsewhere. It could be the arrival of this phytoplankton which triggers shallow diving. This raises a question about why this arrival triggers the opposite response from the onset of a local bloom.

Moreover, the authors present evidence from the literature for other sea ice ecosystems that supports their expectation that phytoplankton blooms are associated with shallower prey distribution. It is therefore interesting to ask why the evidence is not so clear cut in this case. Is it that the chosen metric is a poor indicator of the bloom or is this ecosystem behaving differently? Thus, I think it would be possible for the authors to develop a hypothesis that better fits the evidence. However they would be well advised to avoid repeating the mistake of focusing on the hypothesis rather than the strengths of their data.

I have two more minor comments about language and interpretation:

(1) The metric in Fig 3c is largely a function of dive depth (as travel time is less for shallow dives) and it is inappropriate to describe it as "foraging efficiency". Use a descriptive term similar the one in parentheses in this figure.

(2) This may be a matter of taste but the language around "bottom-up/top-down control" of the food chains implies a set of controls on trophic relationships and/or population dynamics. Thus it seems inappropriate to use this terminology to describe the changes in vertical distribution being discussed, especially as the authors are arguing that feeding relationships are unaffected by these changes.

Additional comments

[37,38] The geographical terms are potentially misleading: "a" rather than "the" Southern Ocean food chain would be more appropriate given the localised scale of the study. Similarly, "the annual phytoplankton bloom in Antarctica" generalises the study to over-simplify the complex dynamics of a vast area. Also "Antarctica" is a continent.

[62] I suggest "concentrates" instead of "aggregates"; delete "and their offspring" which is redundant.

[134] Figure S2?

[207] "trigger"

Fig 3: This would be improved by the addition of sea ice cover and chlorophyll data (as S2).

Decision letter (RSPB-2020-0377.R0)

03-Apr-2020

Dear Dr Beltran:

I am writing to inform you that your manuscript RSPB-2020-0377 entitled "Resource pulses shift the vertical distribution of a Southern Ocean food chain" has, in its current form, been rejected for publication in Proceedings B.

This action has been taken on the advice of referees, who have recommended that substantial revisions are necessary. With this in mind we would be happy to consider a resubmission, provided the comments of the referees are fully addressed. However please note that this is not a provisional acceptance.

The resubmission will be treated as a new manuscript. However, we will approach the same reviewers if they are available and it is deemed appropriate to do so by the Editor. Please note

that resubmissions must be submitted within six months of the date of this email. In exceptional circumstances, extensions may be possible if agreed with the Editorial Office. Manuscripts submitted after this date will be automatically rejected.

Sincerely,
Dr Sasha Dall
mailto: proceedingsb@royalsociety.org

Associate Editor

Comments to Author:

Both reviewers found the study very interesting, well-executed and well-written. A major weakness of the study is, however, that several inferences had to be made that were not directly based on in situ observations. In this context, Reviewer 2 had some major comments particularly about the apparent mismatch in timing between the change in diving depth and the sea-ice break out, hence putative algae bloom. Furthermore, she/he pointed to the need to strengthen parts of the reasoning that are based on indirect information (from other sites). Moreover, this reviewer suggested an alternative hypothesis to explain your data where ocean current velocity is the primary determinant of in situ phytoplankton production timing. This alternative explanation should be explicitly considered. I also agree that terminology such as "bottom-up/top-down control" of the food chains should be avoided given that most of the processes associated with such control have not been explicitly quantified in current study system.

Reviewer(s)' Comments to Author:

Referee: 1

Comments to the Author(s)

This is a well-written and nicely documented paper. Well done. I've no major concerns and my only a handful of minor comments, so let's get to them.

1) It would be helpful to ensure all figures, including supplemental figures, are cited and cited in the order they are discussed in the text. For example - Fig 5a is referred to indirectly only after 5b and 5c are introduced; Figure S1 is not referenced in the text; Fig 3A is not referenced in the text; Figure S3 is referenced before any other.

2) Line 111-112 Are the dates of pupping and molt critical for this analysis? might the end of this sentence be deleted?

3) To help with figure numbering issues noted above and reference Fig 3A (indirectly, at least), I suggest you refer to Figure 3 at the end of the sentence on line 169.

4) Line 190-192. Fair point here, but I feel like a more convincing argument could be made if you

can link to other ancillary evidence from prior tracking studies in the Ross Sea or elsewhere. Do those studies support your assumption that the seals remain reasonably stationary during the period of your tracking.

Line 201 - it may be worth clarifying that foraging efficiency increases because more time can be spent foraging relative to ascent and descent on shallow dives. Presumably total dive time or average dive time did not change much during across the tracking period?

Referee: 2

Comments to the Author(s)

This MS investigates inter-annual differences in the progression of the dive foraging pattern of Weddell Seals and relates these to the timing of ice breakout as a proxy for phytoplankton bloom timing. It also shows, via stable isotope analysis, that the diet of these seals remains relatively constant despite changes in dive depth. This is presented as evidence of a seasonal vertical restructuring of the foodweb in response to the bottom-up effect of the "resource pulse" delivered by the phytoplankton bloom.

I found the paper mainly well written with a clear narrative. I have no concerns about the methodology. It is a perfectly good study of the relationship between diving behaviour and ice-dynamics, which is enhanced by stable isotope data showing that diet remains relatively constant. This supports the interpretation that predator dive distribution tracks prey vertical distribution. This in turn raises some interesting questions about what drives these changes. Although the authors provide convincing evidence that these changes are associated with the timing of ice break-out, their proposed mechanism does not fit the evidence. I also have some concerns about the ambiguity of language around key information about spatial scale and what constitutes a "resource pulse".

The authors have used their observations of predator diving behaviour to infer a sequence of events affecting a whole food chain. I feel that the inference stretches the data beyond its capabilities.

The paper would be stronger if it included data to support these inferences, particularly about the phytoplankton bloom. I acknowledge that the MS provides some information (Fig S2) about the relative timing of sea ice and phytoplankton, but this is at a different location and in a different year from the study. Perhaps more clarity about the relationship between these data and the inferences the authors are making would help.

Unfortunately though, the main conclusions that the "extreme near-surface resource pulse ... resulted in prey shallowing" [40] and "sea ice break-out appears to trigger cascading vertical migrations... shifting control... to bottom up"[195] [also 206], seem to contradict the evidence of Fig 3 which shows that shallow diving (which is presented as an indicator of prey vertical distribution) begins before ice break-out. This shallow diving is described as occurring "for three weeks during the phytoplankton bloom" [198]. Yet "the ice break-out date was considered the approximate start date of in situ primary production" [135]. If anything, break-out seems to mark the beginning of a return to deeper diving.

The ice break-out information is for a location 40km distant from the study site, but there is no discussion of how representative this is of the study site. The data (Fig S2) suggest that peak production follows ice break-out and chlorophyll is lower 5 days before ice break-out than during the post-bloom period two months later (i.e. there is no evidence of a phytoplankton bloom that occurs during the period of shallow diving before ice break-out).

Fig 1 shows that minimum dive depth occurs before 1st January and is followed by a rapid return to deeper diving, whereas Fig S6 shows that ice break-out was always after 1st January. The claim that "At the end of the summer, when... phytoplankton biomass was depleted... seals resumed foraging at (deeper) depths" is not supported by this evidence. It looks like seals begin shallow

foraging before midsummer and return rapidly to deep diving just as the putative bloom develops.

The authors do not explicitly acknowledge this mismatch in timing between observations and the expectation based on their assumptions. The authors allude to the possibility of earlier phytoplankton availability ([135-137], Fig S2 caption) but this is not sufficient to resolve the inconsistency. Also the language around primary production/phytoplankton bloom and timing is ambiguous. Are “in situ primary production” and “phytoplankton bloom” synonymous? How is ice break-out an “estimate of primary production” [136], and what does “conservative” mean in this context? Do you mean it is an estimate of the start of the bloom, but is likely to be late? The authors need to be clear and consistent in describing events that are central to their argument.

Also, the following section of the Supplement lacks any information which allows the reader to relate the sequence of events to the timing metrics (date and ice break-out date) used in the MS. Rather, it suggests that ocean current velocity is the primary determinant of in situ phytoplankton production timing:

“Each summer as sea ice degrades, three sequential processes occur. Initially, sea ice microalgae are released into the water column and can photosynthesize at very low irradiance levels; however, these microalgae contribute little to standing chlorophyll concentration. Next, phytoplankton are advected into the Erebus Bay study area (77.6°S 167.0°E) from the Ross Sea polynya (centered approximately at 77.0°S 175.0°E). About two weeks later (based on ocean current velocity), in situ phytoplankton production markedly increases as nutrients are released and ice no longer shades the photic zone (3).”

So, it seems that the data do not support sea ice break-out as the trigger for deep diving (in contrast to the statements on lines 195 and 206), or an “extreme resource pulse” in the form of a local phytoplankton bloom as the mechanism (in contrast to the central theme of the MS). Rather the data suggest that both of these events mark the end of shallow diving.

The paragraph quoted above suggests that there could be some elevation of phytoplankton at the study site before ice break-out, as a result of algal release elsewhere. It could be the arrival of this phytoplankton which triggers shallow diving. This raises a question about why this arrival triggers the opposite response from the onset of a local bloom.

Moreover, the authors present evidence from the literature for other sea ice ecosystems that supports their expectation that phytoplankton blooms are associated with shallower prey distribution. It is therefore interesting to ask why the evidence is not so clear cut in this case. Is it that the chosen metric is a poor indicator of the bloom or is this ecosystem behaving differently? Thus, I think it would be possible for the authors to develop a hypothesis that better fits the evidence. However they would be well advised to avoid repeating the mistake of focusing on the hypothesis rather than the strengths of their data.

I have two more minor comments about language and interpretation:

- (1) The metric in Fig 3c is largely a function of dive depth (as travel time is less for shallow dives) and it is inappropriate to describe it as “foraging efficiency”. Use a descriptive term similar to the one in parentheses in this figure.
- (2) This may be a matter of taste but the language around “bottom-up/top-down control” of the food chains implies a set of controls on trophic relationships and/or population dynamics. Thus it seems inappropriate to use this terminology to describe the changes in vertical distribution being discussed, especially as the authors are arguing that feeding relationships are unaffected by these changes.

Additional comments

[37,38] The geographical terms are potentially misleading: “a” rather than “the” Southern Ocean food chain would be more appropriate given the localised scale of the study. Similarly, “the annual phytoplankton bloom in Antarctica” generalises the study to over-simplify the complex dynamics of a vast area. Also “Antarctica” is a continent.

[62] I suggest “concentrates” instead of “aggregates”; delete “and their offspring” which is redundant.

[134] Figure S2?

[207] “trigger”

Fig 3: This would be improved by the addition of sea ice cover and chlorophyll data (as S2).

Author's Response to Decision Letter for (RSPB-2020-0377.R0)

See Appendix A.

RSPB-2020-0854.R0

Review form: Reviewer 2

Recommendation

Major revision is needed (please make suggestions in comments)

Scientific importance: Is the manuscript an original and important contribution to its field?

Good

General interest: Is the paper of sufficient general interest?

Good

Quality of the paper: Is the overall quality of the paper suitable?

Marginal

Is the length of the paper justified?

Yes

Should the paper be seen by a specialist statistical reviewer?

No

Do you have any concerns about statistical analyses in this paper? If so, please specify them explicitly in your report.

No

It is a condition of publication that authors make their supporting data, code and materials available - either as supplementary material or hosted in an external repository. Please rate, if applicable, the supporting data on the following criteria.

Is it accessible?

Yes

Is it clear?

Yes

Is it adequate?

No

Do you have any ethical concerns with this paper?

No

Comments to the Author

Thank you to the authors for taking some of my comments on board.

The manuscript has improved and the revised figure 3 helps to clarify the relationships between seal behaviour, the timing of ice break-out at McMurdo sound and putative phytoplankton availability (but the main text needs to provide a clear and referenced justification for using this temporally shifted phytoplankton data from elsewhere – see comments on para beginning line 127 and Fig 3). As in my previous comments, a critical feature of these relationships is that ice break-out occurs AFTER peak phytoplankton availability. Unfortunately the paper is still confused about this point and it argues against logic that ice break-out is an “early estimate for the date of peak primary production” (line 131) and that it “drives” the observed behaviour (line 150).

To be frank, the paper is built round a correlative relationship which is presented as causal but which clearly can not be causal.

There is a coherent explanation of the results which can be constructed from the information provided (see comments on para beginning line 127) as follows: the TIMING OF ICE BREAK-OUT at McMurdo Sound is an indicator of large scale processes that drive the timing of seasonal events across this part of the Southern Ocean. These include events which occur earlier in the season such as the timing of the phytoplankton bloom at Cape Crozier. It is likely that ADVECTIVE ARRIVAL OF MATERIAL from this bloom at the study site causes a resource pulse at the study site which drives changes in the distribution of prey and hence the foraging behaviour of Weddell Seals. While IN SITU PRODUCTION at the study site might increase after ice break-out, it seems insufficient to prevent the prey from returning to deeper waters. The narrative offered by the authors blurs the boundaries between the three processes highlighted in upper case above and uses the phrases “phytoplankton boom” and “primary production” to describe any or all of them. In fact the new edits include changing the phrase “ice break-out” to “phytoplankton bloom” (line 153) even though the authors argue “the actual bloom may precede the date of ice breakout” (response to reviews) and, in the same paragraph (and line 131), that ice break-out is an “early estimate for the date of peak primary production”. So the authors seem to be claiming that ice break-out is simultaneously equivalent to, after, and before the resource pulse!

To improve the manuscript based on the current analysis, the authors must:

- (1) Recognise that local ice break-out may be an INDICATOR of timing but can not be a DRIVER of events that precede it.
- (2) Be clear in the main text about the reasoning linking their data (TIMING OF ICE BREAK-OUT) to the processes it putatively indicates (ADVECTIVE ARRIVAL OF MATERIAL, IN SITU PRODUCTION) and the evidence that supports this.
- (3) Use language that clearly distinguishes between the three processes. I concede the authors’ point that “Phytoplankton bloom usually refers to a high concentration of phytoplankton in an area regardless of where the production originated” but I find “resource pulse” a much better term to describe the combined effect of ADVECTIVE ARRIVAL OF MATERIAL and any co-occurring IN SITU PRODUCTION. My advice is to use one phrase consistently and to define it at first use. ADVECTIVE ARRIVAL OF MATERIAL should not be described as (peak) (primary) production since the production happened in the past and elsewhere.
- (4) Be clear (in the paper itself rather than the supplement) about the spatial relationship between their study site, the location of the ICE BREAK-OUT data and the location where the advected material originates.

With more clarity about the sequence of events and the differences between drivers and indicators the study might be publishable without any further analysis. It would be more

complete (and therefore better suited to a high impact journal) if the authors were able to test the hypothesis that ADVECTIVE ARRIVAL OF MATERIAL is more important than local IN SITU PRODUCTION. This could be done by constructing alternative GAMMs using a direct indicator of bloom timing at Cape Crozier as an explanatory variable. Does this explanatory variable perform better than TIMING OF ICE BREAK-OUT in McMurdo sound? Is the direct indicator strongly correlated with TIMING OF ICE BREAK-OUT in McMurdo sound?

As part of the review process I was asked to check data availability at <http://www.usap-dc.org/view/dataset/601137>. The data fields available are Days since 1 January, Daily mean dive depth and Season. The data required to plot Figs 3C, 3D, 4 and 5 do not seem to be available.

38 - "deep waters" is misleading; "at depths \geq X m" would be better, or "at depth" might work.

59 - I suggest "in" rather than "along" the marginal ice zone.

64-65 - I suggest continuing the penultimate sentence with "and it is therefore important to understand how ecological dynamics vary across time and 3-dimensional space.", and ending the paragraph there. There is no intervention that will fix (address) this specific issue.

68 - replace "depth-driven clines" with "depth gradients" or similar. This is to remove the debatable assertion that depth is the main driver of all of these variables.

74 - add "increasing" before "depth".

78 - I suggest deleting the following: "which allows them to capitalize on the proximity of their food and oxygen supplies and thus optimize foraging efficiency." This is an unnecessary level of subjective interpretation of the fact that predators forage where there are prey.

82 - I suggest "use of vertical space"

83 - The phrase "the Southern Ocean's most productive region" implies a relevance which isn't demonstrated. The Ross Sea sector may be the Southern Ocean's "most productive region" based on averages over millions of square km but that doesn't mean that your Weddell seal foraging site is more productive than sites in other sectors. Either delete the phrase or rephrase to clarify the difference in scales, along the lines, "at a study site in the Ross Sea, which is in the most productive sector of the Southern Ocean"

127 - This paragraph is still not providing all of the information required to understand ice break-out date as a proxy for resource pulse timing at the study site. Please make sure it contains all of the following info (gleaned mainly from your supplementary information): (1) You do not have direct observations of phytoplankton availability at the study site; (2) "As sea ice extent decreases phytoplankton concentration increases via in situ production"; (3) "After the sea ice breaks out, in situ phytoplankton production markedly increases as nutrients are released and ice no longer shades the photic zone"; (4) This (enhanced in situ production following break out) typically occurs 54 days earlier at Cape Crozier than the study site, the resultant phytoplankton is advected to the study site in approximately 23 days and therefore the "resource pulse" at the study site is likely to start about 31 days before ice break-out; (5) Ice-break out supplements this resource pulse by catalysing a "marked increase" in in situ production at the study site; (6) but "Advective inputs are thought to contribute significantly more to water column productivity than in situ production". All of these assertions require supporting references.

131 - "The ice break-out date is an early estimate for the date of peak primary production". Surely it is a late estimate. Your fig 3 implies the peak is between 30 and 10 days BEFORE ice break-out.

150-158 – This paragraph is about general patterns and does not mention differences between years, so it should refer to Fig 3 rather than Fig 1. This would also allow the authors to put specific dates relative to ice break-out on the various statements.

153 – The phrase “phytoplankton bloom” is used here to replace “ice break-out” (in the earlier version of the MS). I prefer the latter which is the tangible variable analysed, as opposed to the former which is difficult to pin-point in time, but which doesn’t seem to be coincident with ice break-out. However I suggest deleting this whole sentence and replacing it with a statement about the period ending about 31 days before ice break-out which seems to be a reasonable definition of before the bloom.

160-165 – Again I don’t support the use of the phrase “phytoplankton bloom” as a synonym for “ice break-out”, please revert to the original.

Fig 3 – this needs a clear statement that the chlorophyll data are from Cape Crozier but are shifted by 23 days on the x-axis to represent transport time to the study site. The justification for this, including supporting references, should be in the main text. The figure could be improved further with a representation of in-situ production at the study site (the text suggests three key issues to capture: In situ production starts before break-out, it increases at break-out, but it contributes less to the resource pulse than advected material). Given the lack of data, this would have to be diagrammatic.

Fig S4. This is useful. The map could also locate the study site and location of the ice break out data. I suggest labeling Ross Island. Remove the word “dashed” as the lines seem solid.

Decision letter (RSPB-2020-0854.R0)

12-Jun-2020

Dear Dr Beltran:

Your manuscript has now been peer reviewed and the reviews have been assessed by an Associate Editor. The reviewers’ comments (not including confidential comments to the Editor) and the comments from the Associate Editor are included at the end of this email for your reference. As you will see, the reviewers and the Editors have raised some concerns with your manuscript and we would like to invite you to revise your manuscript to address them.

When submitting your revision please upload a file under "Response to Referees" in the "File Upload" section. This should document, point by point, how you have responded to the reviewers’ and Editors’ comments, and the adjustments you have made to the manuscript. We require a copy of the manuscript with revisions made since the previous version marked as ‘tracked changes’ to be included in the ‘response to referees’ document.

Research ethics:

Use of animals and field studies:

Please submit a copy of your revised paper within three weeks. If we do not hear from you within this time your manuscript will be rejected. If you are unable to meet this deadline please let us know as soon as possible, as we may be able to grant a short extension.

Best wishes,
Dr Sasha Dall
mailto: proceedingsb@royalsociety.org

Associate Editor Board Member

Comments to Author:

While important improvements were made in this resubmission, the paper still needs considerable improvement to reach the level of excellence expected for PRSB. The major comment of the reviewer remains that your key pattern is based on a correlative relationship between ice break-out and phytoplankton availability which is presented as causal, and moreover still lacks clarity. The reviewer makes excellent suggestions how to rephrase the sequence of events and the differences between drivers and indicators generating the observed patterns. Importantly, to make your story more convincing the suggested analysis to test the hypothesis that advective arrival of material is more important than local in situ production will be pivotal to improve your study.

Reviewer(s)' Comments to Author:

Referee: 2

Comments to the Author(s).

Thank you to the authors for taking some of my comments on board.

The manuscript has improved and the revised figure 3 helps to clarify the relationships between seal behaviour, the timing of ice break-out at McMurdo sound and putative phytoplankton availability (but the main text needs to provide a clear and referenced justification for using this temporally shifted phytoplankton data from elsewhere – see comments on para beginning line 127 and Fig 3). As in my previous comments, a critical feature of these relationships is that ice break-out occurs AFTER peak phytoplankton availability. Unfortunately the paper is still confused about this point and it argues against logic that ice break-out is an “early estimate for the date of peak primary production” (line 131) and that it “drives” the observed behaviour (line 150).

To be frank, the paper is built round a correlative relationship which is presented as causal but which clearly can not be causal.

There is a coherent explanation of the results which can be constructed from the information provided (see comments on para beginning line 127) as follows: the TIMING OF ICE BREAK-OUT at McMurdo Sound is an indicator of large scale processes that drive the timing of seasonal events across this part of the Southern Ocean. These include events which occur earlier in the season such as the timing of the phytoplankton bloom at Cape Crozier. It is likely that ADVECTIVE ARRIVAL OF MATERIAL from this bloom at the study site causes a resource pulse at the study site which drives changes in the distribution of prey and hence the foraging behaviour of Weddell Seals. While IN SITU PRODUCTION at the study site might increase after ice break-out, it seems insufficient to prevent the prey from returning to deeper waters.

The narrative offered by the authors blurs the boundaries between the three processes highlighted in upper case above and uses the phrases “phytoplankton boom” and “primary production” to describe any or all of them. In fact the new edits include changing the phrase “ice break-out” to “phytoplankton bloom” (line 153) even though the authors argue “the actual bloom may precede the date of ice breakout” (response to reviews) and, in the same paragraph (and line 131), that ice break-out is an “early estimate for the date of peak primary production”. So the authors seem to be claiming that ice break-out is simultaneously equivalent to, after, and before the resource pulse!

To improve the manuscript based on the current analysis, the authors must:

(1) Recognise that local ice break-out may be an INDICATOR of timing but can not be a DRIVER of events that precede it.

(2) Be clear in the main text about the reasoning linking their data (TIMING OF ICE BREAK-OUT) to the processes it putatively indicates (ADVECTIVE ARRIVAL OF MATERIAL, IN SITU PRODUCTION) and the evidence that supports this.

(3) Use language that clearly distinguishes between the three processes. I concede the authors' point that "Phytoplankton bloom usually refers to a high concentration of phytoplankton in an area regardless of where the production originated" but I find "resource pulse" a much better term to describe the combined effect of ADVECTIVE ARRIVAL OF MATERIAL and any co-occurring IN SITU PRODUCTION. My advice is to use one phrase consistently and to define it at first use. ADVECTIVE ARRIVAL OF MATERIAL should not be described as (peak) (primary) production since the production happened in the past and elsewhere.

(4) Be clear (in the paper itself rather than the supplement) about the spatial relationship between their study site, the location of the ICE BREAK-OUT data and the location where the advected material originates.

With more clarity about the sequence of events and the differences between drivers and indicators the study might be publishable without any further analysis. It would be more complete (and therefore better suited to a high impact journal) if the authors were able to test the hypothesis that ADVECTIVE ARRIVAL OF MATERIAL is more important than local IN SITU PRODUCTION. This could be done by constructing alternative GAMMs using a direct indicator of bloom timing at Cape Crozier as an explanatory variable. Does this explanatory variable perform better than TIMING OF ICE BREAK-OUT in McMurdo sound? Is the direct indicator strongly correlated with TIMING OF ICE BREAK-OUT in McMurdo sound?

As part of the review process I was asked to check data availability at <http://www.usap-dc.org/view/dataset/601137>. The data fields available are Days since 1 January, Daily mean dive depth and Season. The data required to plot Figs 3C, 3D, 4 and 5 do not seem to be available.

38 - "deep waters" is misleading; "at depths > X_m" would be better, or "at depth" might work.

59 - I suggest "in" rather than "along" the marginal ice zone.

64-65 - I suggest continuing the penultimate sentence with "and it is therefore important to understand how ecological dynamics vary across time and 3-dimensional space.", and ending the paragraph there. There is no intervention that will fix (address) this specific issue.

68 - replace "depth-driven clines" with "depth gradients" or similar. This is to remove the debatable assertion that depth is the main driver of all of these variables.

74 - add "increasing" before "depth".

78 - I suggest deleting the following: "which allows them to capitalize on the proximity of their food and oxygen supplies and thus optimize foraging efficiency." This is an unnecessary level of subjective interpretation of the fact that predators forage where there are prey.

82 - I suggest "use of vertical space"

83 - The phrase "the Southern Ocean's most productive region" implies a relevance which isn't demonstrated. The Ross Sea sector may be the Southern Ocean's "most productive region" based on averages over millions of square km but that doesn't mean that your Weddell seal foraging site is more productive than sites in other sectors. Either delete the phrase or rephrase to clarify the difference in scales, along the lines, "at a study site in the Ross Sea, which is in the most productive sector of the Southern Ocean"

127 - This paragraph is still not providing all of the information required to understand ice break-out date as a proxy for resource pulse timing at the study site. Please make sure it contains all of

the following info (gleaned mainly from your supplementary information): (1) You do not have direct observations of phytoplankton availability at the study site; (2) “As sea ice extent decreases phytoplankton concentration increases via in situ production”; (3) “After the sea ice breaks out, in situ phytoplankton production markedly increases as nutrients are released and ice no longer shades the photic zone”; (4) This (enhanced in situ production following break out) typically occurs 54 days earlier at Cape Crozier than the study site, the resultant phytoplankton is advected to the study site in approximately 23 days and therefore the “resource pulse” at the study site is likely to start about 31 days before ice break-out; (5) Ice-break out supplements this resource pulse by catalysing a “marked increase” in in situ production at the study site; (6) but “Advective inputs are thought to contribute significantly more to water column productivity than in situ production”. All of these assertions require supporting references.

131 - “The ice break-out date is an early estimate for the date of peak primary production”. Surely it is a late estimate. Your fig 3 implies the peak is between 30 and 10 days BEFORE ice break-out.

150-158 - This paragraph is about general patterns and does not mention differences between years, so it should refer to Fig 3 rather than Fig 1. This would also allow the authors to put specific dates relative to ice break-out on the various statements.

153 - The phrase “phytoplankton bloom” is used here to replace “ice break-out” (in the earlier version of the MS). I prefer the latter which is the tangible variable analysed, as opposed to the former which is difficult to pin-point in time, but which doesn’t seem to be coincident with ice break-out. However I suggest deleting this whole sentence and replacing it with a statement about the period ending about 31 days before ice break-out which seems to be a reasonable definition of before the bloom.

160-165 - Again I don’t support the use of the phrase “phytoplankton bloom” as a synonym for “ice break-out”, please revert to the original.

Fig 3 - this needs a clear statement that the chlorophyll data are from Cape Crozier but are shifted by 23 days on the x-axis to represent transport time to the study site. The justification for this, including supporting references, should be in the main text. The figure could be improved further with a representation of in-situ production at the study site (the text suggests three key issues to capture: In situ production starts before break-out, it increases at break-out, but it contributes less to the resource pulse than advected material). Given the lack of data, this would have to be diagrammatic.

Fig S4. This is useful. The map could also locate the study site and location of the ice break out data. I suggest labeling Ross Island. Remove the word “dashed” as the lines seem solid.

Author's Response to Decision Letter for (RSPB-2020-0854.R0)

See Appendix B.

RSPB-2020-0854.R1 (Revision)

Review form: Reviewer 2

Recommendation

Major revision is needed (please make suggestions in comments)

Scientific importance: Is the manuscript an original and important contribution to its field?
Good

General interest: Is the paper of sufficient general interest?
Good

Quality of the paper: Is the overall quality of the paper suitable?
Marginal

Is the length of the paper justified?
Yes

Should the paper be seen by a specialist statistical reviewer?
No

Do you have any concerns about statistical analyses in this paper? If so, please specify them explicitly in your report.
No

It is a condition of publication that authors make their supporting data, code and materials available - either as supplementary material or hosted in an external repository. Please rate, if applicable, the supporting data on the following criteria.

Is it accessible?
N/A

Is it clear?
N/A

Is it adequate?
N/A

Do you have any ethical concerns with this paper?
No

Comments to the Author

Line numbers refer to the tracked changes version of the MS.

I have reviewed earlier versions of this MS. I advised in both reviews that (1) it is essential to be clear and consistent about central concepts in the narrative, including the timing and duration of the resource pulse that putatively drives seal behaviour; and (2) it is not scientifically valid to present ice break-out at the study site as a driver of events that precede it.

The Methods of the current MS define the “resource pulse” as “the combined effect of the advective arrival of phytoplankton from the Cape Crozier phytoplankton bloom and the local in situ production” (lines 164 to 166), i.e. a protracted process starting approx. 31 days before, and ending some time after, ice break-out. Yet in the Results and figures it is treated as a specific date (i.e. a process lasting no longer than a day) (e.g. lines 183-201), fig caption 3, figs 2, 3, 6). This leads to particular confusion with lines 187 to 191. The first statement (187) suggests that shallowing is coincident with advection AND in situ production while the second (190) suggests that this PRECEDES a return to deeper diving. Yet the return begins before the event (ice break-out) (S2) which catlayzes “a marked increase in in situ production at the study site” (line 162).

The abstract suggests that ice break-out is a driver of behaviour “with later ice break-out delaying periods of seal dive shallowing by one month.” (line 45).

I further suggested that “It would be more complete (and therefore better suited to a high impact journal) if the authors were able to test the hypothesis that ADVECTIVE ARRIVAL OF MATERIAL is more important than local IN SITU PRODUCTION. This could be done by constructing alternative GAMMs using a direct indicator of bloom timing at Cape Crozier as an explanatory variable.”

The authors have not done this, claiming that “the data are not available”. To clarify, my suggestion was to use ice break-out timing at Cape Crozier as an indicator of bloom timing at that site. I find it surprising that NASA Bootstrap Sea Ice Concentrations are not available for Cape Crozier but are available for McMurdo.

Given that the current revision falls short on these three critical points, I do not think it is ready for publication.

To reiterate: This is a valid and interesting study. The analyses have merit. However the “explanatory variable” (local ice break-out date) is, at best, a proxy, for events that predate it. If local in situ production is highest after ice break out, then it neither causes nor maintains shallow diving. The authors could consider a potentially more relevant explanatory variable (Cape Crozier ice break out). Nonetheless the onus is on them to provide a clear and consistent explanation of their findings which is also clear about the limitations of their data. I do not think that a series of hasty revisions is the best way to achieve this and I strongly advise that the authors make their own efforts to review the MS for clarity and consistency before the next submission to a journal.

Decision letter (RSPB-2020-0854.R1)

01-Sep-2020

Dear Dr Beltran:

I am writing to inform you that your manuscript # RSPB-2020-0854.R1 entitled "Resource pulses shift the vertical distribution of a Southern Ocean food chain" has been rejected for publication in Proceedings B.

This action has been taken on the advice of referees, who have recommended that substantial revisions are necessary. With this in mind we would be happy to consider a resubmission, provided the comments of the referees are fully addressed. However please note that this is not a provisional acceptance.

Please find below the comments made by the referees, not including confidential reports to the Editor, which I hope you will find useful.

- 1) A 'response to referees' document including details of how you have responded to the comments, and the adjustments you have made.
- 2) A clean copy of the manuscript and one with 'tracked changes' indicating your 'response to referees' comments document.
- 3) Line numbers in your main document.
- 4) Please read our Data sharing policies to ensure that you meet our requirements (<https://royalsociety.org/journals/authors/author-guidelines/#data>).

Sincerely,
 Dr Sasha Dall
 Editor, Proceedings B
proceedingsb@royalsociety.org

Reviewer(s)' Comments to Author:

Referee: 2

Comments to the Author(s)

Line numbers refer to the tracked changes version of the MS.

I have reviewed earlier versions of this MS. I advised in both reviews that (1) it is essential to be clear and consistent about central concepts in the narrative, including the timing and duration of the resource pulse that putatively drives seal behaviour; and (2) it is not scientifically valid to present ice break-out at the study site as a driver of events that precede it.

The Methods of the current MS define the "resource pulse" as "the combined effect of the advective arrival of phytoplankton from the Cape Crozier phytoplankton bloom and the local in situ production" (lines 164 to 166), i.e. a protracted process starting approx. 31 days before, and ending some time after, ice break-out. Yet in the Results and figures it is treated as a specific date (i.e. a process lasting no longer than a day) (e.g. lines 183-201), fig caption 3, figs 2, 3, 6). This leads to particular confusion with lines 187 to 191. The first statement (187) suggests that shallowing is coincident with advection AND in situ production while the second (190) suggests that this PRECEDES a return to deeper diving. Yet the return begins before the event (ice break-out) (S2) which catlayses "a marked increase in in situ production at the study site" (line 162).

The abstract suggests that ice break-out is a driver of behaviour "with later ice break-out delaying periods of seal dive shallowing by one month." (line 45).

I further suggested that "It would be more complete (and therefore better suited to a high impact journal) if the authors were able to test the hypothesis that ADVECTIVE ARRIVAL OF MATERIAL is more important than local IN SITU PRODUCTION. This could be done by constructing alternative GAMMs using a direct indicator of bloom timing at Cape Crozier as an explanatory variable."

The authors have not done this, claiming that "the data are not available". To clarify, my suggestion was to use ice break-out timing at Cape Crozier as an indicator of bloom timing at that site. I find it surprising that NASA Bootstrap Sea Ice Concentrations are not available for Cape Crozier but are available for McMurdo.

Given that the current revision falls short on these three critical points, I do not think it is ready for publication.

To reiterate: This is a valid and interesting study. The analyses have merit. However the “explanatory variable” (local ice break-out date) is, at best, a proxy, for events that predate it. If local in situ production is highest after ice break out, then in neither causes nor maintains shallow diving. The authors could consider a potentially more relevant explanatory variable (Cape Crozier ice break out). Nonetheless the onus is on them to provide a clear and consistent explanation of their findings which is also clear about the limitations of their data. I do not think that a series of hasty revisions is the best way to achieve this and I strongly advise that the authors make their own efforts to review the MS for clarity and consistency before the next submission to a journal.

Author's Response to Decision Letter for (RSPB-2020-0854.R1)

See Appendix C.

RSPB-2020-2817.R0

Review form: Reviewer 1

Recommendation

Accept with minor revision (please list in comments)

Scientific importance: Is the manuscript an original and important contribution to its field?

Good

General interest: Is the paper of sufficient general interest?

Good

Quality of the paper: Is the overall quality of the paper suitable?

Good

Is the length of the paper justified?

Yes

Should the paper be seen by a specialist statistical reviewer?

No

Do you have any concerns about statistical analyses in this paper? If so, please specify them explicitly in your report.

No

It is a condition of publication that authors make their supporting data, code and materials available - either as supplementary material or hosted in an external repository. Please rate, if applicable, the supporting data on the following criteria.

Is it accessible?

Yes

Is it clear?

Yes

Is it adequate?

Yes

Do you have any ethical concerns with this paper?

No

Comments to the Author

Dear Authors,

I reviewed an earlier version of this manuscript, which was largely acceptable to me at that time. Nonetheless, it has improved in the interim and now offers a more nuanced and cautious view of the role of ice-breakout and subsequent phytoplankton blooms on diving behavior of Weddell seals. In general, I think you have done a decent job adapting your analyses to the requests of reviewers and I see the current manuscript as acceptable for publication, noting a handful a few minor suggestions for you to consider:

Line 208 - should be "proceeded" or "progressed"

Line 261 - for the sake of clarity, please specify which characteristic of the ice station most closely matches seal shallowing.

Lines 217-229 - for each specific result mentioned, it would be very helpful to reference the figure (and panel) that presents the data/result. For example, the first sentence should reference figure 2. On a related note, the map presented in figure 3 might be better housed if combined as a second panel for figure 2 to show the spatial and temporal elements of resource pulsing in one coherent figure.

Line 219-222: While this may be a true statement according to the table in figure 3, it would be more helpful to highlight results for covariates that best explain the pattern in the data, rather than focus on one that doesn't (or, why would we expect Julian day to be a good predictor of a behavior so intimately tied to natural variability in the environment?)

Review form: Reviewer 3

Recommendation

Major revision is needed (please make suggestions in comments)

Scientific importance: Is the manuscript an original and important contribution to its field?

Excellent

General interest: Is the paper of sufficient general interest?

Excellent

Quality of the paper: Is the overall quality of the paper suitable?

Acceptable

Is the length of the paper justified?

Yes

Should the paper be seen by a specialist statistical reviewer?

No

Do you have any concerns about statistical analyses in this paper? If so, please specify them explicitly in your report.

No

It is a condition of publication that authors make their supporting data, code and materials available - either as supplementary material or hosted in an external repository. Please rate, if applicable, the supporting data on the following criteria.

Is it accessible?

Yes

Is it clear?

Yes

Is it adequate?

Yes

Do you have any ethical concerns with this paper?

No

Comments to the Author

This interesting paper integrates measures of seal foraging behaviour with oceanographic data to test hypotheses about how the arrival of resource subsidies (advected phytoplankton) cause the vertical redistribution in the water column of multiple trophic levels. The vertical movement of only one trophic level, seals, is measured directly, while that of fish (the seals' prey) and of zooplankton (the fish's prey) is assumed based on the literature. This revision has benefitted from a prior review. I find that the manuscript contains rigorous analyses that should be of general interest to ecologists, oceanographers, and others interested in the ecological implications of global environmental change. However, several statements and inferences are unclear or not well supported, which made me – an interested reader sympathetic to the many complexities inherent to field data and analyses – struggle to discern the unifying themes and main points of the study. This issue can be solved with careful re-writing.

My view is that the manuscript would greatly improve if the authors would actively accept, from the outset, that support is weak to inconsistent for the hypothesis that shallowing is triggered by the arrival of advected phytoplankton. The authors rigorously tested that hypothesis, which is no small feat, yet the data lead to the conclusion that (lines 261-262) “there is a delay between seal shallowing and the resource pulse that cannot be explained by our existing understanding of the marine ecosystem.” A revision should highlight evidence for that delay, rather than muddling the narrative by struggling to find evidence for synchrony between seal behaviour and the arrival of subsidies, as the current version does.

For instance, the abstract, states that “dive shallowing often coincided with annual ice break-out and advection of the phytoplankton bloom from the surrounding areas,” but this is ambiguous. Instead, the abstract might highlight that, averaged across all years, the shallowing of seal dives began ≈ 25 to ≈ 60 days before the arrival of advected phytoplankton (depending on subsidy source: Fig 3, lower right panel).

Similarly, lines 217-221 of results describe temporal and spatial variability in the data, and the extent to which it might support a correlation between the shallowing of seal dives and the arrival of advected phytoplankton, and states that “depth was better explained by the date of advected phytoplankton arrival than calendar date [except for Northeast]” which, though true in relative sense, it detracts from the stronger evidence for resource pulse arrivals lagging behind seal shallowing (lower right panel of Fig 3, Fig 4 and the lower panel of Fig 6). Why not just streamline the text to highlight those lags, as current lines 224-227 do?

These issues carry on to the discussion, which states (lines 247-249) that “The inter- and intra-annual seal shallowing only roughly coincided with estimates of advected phytoplankton from the stations, suggesting that the phenology of these key processes is a complex issue.” This ambiguous statement should be deleted altogether, and instead the section should lead with the key conclusion from lines 261-262: “there is a delay between seal shallowing and the resource pulse that cannot be explained by our existing understanding of the marine ecosystem.” This lag is an important finding. As the discussion already examines, the lag may be influenced by oceanographic variables that either are not being accounted for or poorly represented. Perhaps it is also worth exploring hypothesis derived from a game theoretic perspective (e.g. reference 1), in which the prey of seals might be avoiding peak resource aggregations to optimize risk-energy tradeoffs, which in turn might contribute to an apparent mismatch between seal shallowing and the arrival of advected phytoplankton.

Perhaps my concern is epitomized by Figure 6. The lower panel shows data that clearly indicate that seal shallowing begins and ends before the arrival of advected phytoplankton, yet those shallow dives are lined up with the portion of the upper panel that represents the expected duration of the resource pulse. Yet expectation and data did not match, and this figure muddles that. Having said that, the intended concept of the figure is great; it should be easy to redraw it for the upper panel to be consistent with the data.

Analyses that disentangle the relative contribution of ice break-out from that of advection of subsidies are a strength of the paper. But I also find that the writing should improve for that part of the narrative. For instance, in their response letter to the earlier reviewer, the authors state that “Advection is faster than break-out, which could explain why local break-out occurs substantially later than seal dive shallowing.” This important statement should be worked into the abstract and discussion.

Other comments

Lines 76-78 (Introduction) state that “Behavioral theory suggests that intermediate trophic levels such as zooplankton and fishes will maximize fitness by balancing ecological trade-offs between resource acquisition and predation risk, which both decrease with increasing depth.” This perspective on the tradeoffs inherent to depth choice is too narrow; cases in which both risk and potential energy gain increase at greater depth are entirely plausible (e.g. reference 2).

Lines 92-97 (Introduction) state that: “We hypothesized that during spring, when resource stratification was weak, predation risk would control the food chain distribution, and intermediate trophic levels (zooplankton and fish) would have deeper distributions. However, we anticipated that a strong resource pulse at the ocean surface during summer would cause these consumers to utilize shallower habitat because the benefits of consuming the abundant resource would out-weigh the risk of predation from air-breathing predators at the surface.” The authors did not test this hypothesis, as they have no data on zooplankton and fish vertical distributions. Rather, their statement is a potential interpretation of results better suited for the discussion.

Lines 124-125 (Methods): “...defined the date of sea ice break-out as the first occurrence of a 7-day running mean ice concentration of $<50\%$ at each gridded data location.” Specify here the size of grid cells (25x25km?)

Lines 125-129 (Methods): Are the coordinates the center points of grid cells?

Lines 145 (Methods): Is the sample size ($n=2,941$) the number of seal days recorded? Clarify.

Lines 174-176 (Results): “Yearly variation in the timing of shallow seal diving also appeared to alter the depth distribution of lower trophic levels, because earlier ice break-out and the concomitantly earlier resource pulse led to shallower seal diving.” This sentence is conceptually muddled because you show no data on “the depth distribution of lower trophic levels.”

Lines 176-177 of results state that “earlier ice break-out and the concomitantly earlier resource pulse led to shallower seal diving” and then references Fig. 1, but that figure has no data on ice break-out or resource pulse. The lower panel of Figure 1 (plot of depth of shallowest dive by date of shallowest dive) is not particularly informative and would more directly address the authors point if its X-axis represented dates related to ice breakout or subsidy arrivals.

Lines 234-237 of the discussion state that “Foraging depth of Weddell seals halved and remained shallow for three weeks in mid-summer, suggesting that zooplankton and fishes had shallowed to take advantage of the new phytoplankton availability, despite the predation risk from air-breathing predators.” But seals are deepening their dives before the arrival of phytoplankton subsidies! Therefore, the data cannot support that statement. Should this be reframed as a hypothesis based on in situ production of phytoplankton? Or is that implausible because “local break-out occurs substantially later than seal dive shallowing”?

Minor comments

Line 208. Apparent typo: “proceeded” vs “preceded”

Lines 237-239. There is a wealth of literature supporting the notion that bottom time (and, potentially, net energy gain) increases with the shallowing of dives. You might consider citing one of those papers here.

Figs 2 and 3. The term “ice” is attached to each location name on the text embedded in the figure (And table in Fig3). This is confusing, as the locations are associated with the timing of both ice break-out and advection. Is the term “ice” really meant to be there?

Fig 3 The map should include labels for Cape Royds, Erebus Bay, Ross Island

Fig 5. Caption should specify that each line in panels B and C represents an individual seal.

References

1. Lima, S. L. Putting predators back into behavioral predator-prey interactions. *Trends Ecol. Evol.* 17, 70-75 (2002).
2. Frid, A., Burns, J., Baker, G. G. & Thorne, R. E. Predicting synergistic effects of resources and predators on foraging decisions by juvenile Steller sea lions. *Oecologia* 158, (2009).

Decision letter (RSPB-2020-2817.R0)

29-Dec-2020

Dear Dr Beltran:

Your manuscript has now been peer reviewed and the reviews have been assessed by an Associate Editor. The reviewers’ comments (not including confidential comments to the Editor) and the comments from the Associate Editor are included at the end of this email for your reference. As you will see, the reviewers and the Editors have raised some concerns with your manuscript and we would like to invite you to revise your manuscript to address them.

Research ethics:

Use of animals and field studies:

It is a condition of publication that you make available the data and research materials supporting the results in the article (<https://royalsociety.org/journals/authors/author-guidelines/#data>). Datasets should be deposited in an appropriate publicly available repository and details of the associated accession number, link or DOI to the datasets must be included in the Data Accessibility section of the article (<https://royalsociety.org/journals/ethics-policies/data-sharing-mining/>). Reference(s) to datasets should also be included in the reference list of the article with DOIs (where available).

Please submit a copy of your revised paper within three weeks. If we do not hear from you within this time your manuscript will be rejected. If you are unable to meet this deadline please let us know as soon as possible, as we may be able to grant a short extension.

Best wishes,
Dr Sasha Dall
mailto:proceedingsb@royalsociety.org

Associate Editor Board Member

Comments to Author:

The authors considerably improved their revised paper. Nevertheless, I agree with the new Reviewer 2 that still too much ambiguity remains about support for the main conclusion of a direct coupling between ice-breakout, the arrival of advected phytoplankton and the shallowing of diving behavior of Weddell seals. A temporal decoupling of these phenomena precludes making strong conclusions thereby repeating a major comment already made during the first review round. A further, careful re-writing will be needed to avoid over-interpretation and ambiguity which will be crucial to make this paper acceptable. I also want to note that the title gives the wrong impression that data were collected on the entire food chain, which is overselling the study.

Reviewer(s)' Comments to Author:

Referee: 1

Comments to the Author(s).

Dear Authors,

I reviewed an earlier version of this manuscript, which was largely acceptable to me at that time. Nonetheless, it has improved in the interim and now offers a more nuanced and cautious view of the role of ice-breakout and subsequent phytoplankton blooms on diving behavior of Weddell seals. In general, I think you have done a decent job adapting your analyses to the requests of reviewers and I see the current manuscript as acceptable for publication, noting a handful a few minor suggestions for you to consider:

Line 208 – should be “proceeded” or “progressed”

Line 261 – for the sake of clarity, please specify which characteristic of the ice station most closely matches seal shallowing.

Lines 217-229 - for each specific result mentioned, it would be very helpful to reference the figure (and panel) that presents the data/result. For example, the first sentence should reference figure 2. On a related note, the map presented in figure 3 might be better housed if combined as a second panel for figure 2 to show the spatial and temporal elements of resource pulsing in one coherent figure.

Line 219-222: While this may be a true statement according to the table in figure 3, it would be more helpful to highlight results for covariates that best explain the pattern in the data, rather than focus on one that doesn't (or, why would we expect Julian day to be a good predictor of a behavior so intimately tied to natural variability in the environment?)

Referee: 3

Comments to the Author(s).

This interesting paper integrates measures of seal foraging behaviour with oceanographic data to test hypotheses about how the arrival of resource subsidies (advected phytoplankton) cause the vertical redistribution in the water column of multiple trophic levels. The vertical movement of only one trophic level, seals, is measured directly, while that of fish (the seals' prey) and of zooplankton (the fish's prey) is assumed based on the literature. This revision has benefitted from a prior review. I find that the manuscript contains rigorous analyses that should be of general interest to ecologists, oceanographers, and others interested in the ecological implications of global environmental change. However, several statements and inferences are unclear or not well supported, which made me – an interested reader sympathetic to the many complexities inherent to field data and analyses – struggle to discern the unifying themes and main points of the study. This issue can be solved with careful re-writing.

My view is that the manuscript would greatly improve if the authors would actively accept, from the outset, that support is weak to inconsistent for the hypothesis that shallowing is triggered by the arrival of advected phytoplankton. The authors rigorously tested that hypothesis, which is no small feat, yet the data lead to the conclusion that (lines 261-262) “there is a delay between seal shallowing and the resource pulse that cannot be explained by our existing understanding of the marine ecosystem.” A revision should highlight evidence for that delay, rather than muddling the narrative by struggling to find evidence for synchrony between seal behaviour and the arrival of subsidies, as the current version does.

For instance, the abstract, states that “dive shallowing often coincided with annual ice break-out and advection of the phytoplankton bloom from the surrounding areas,” but this is ambiguous. Instead, the abstract might highlight that, averaged across all years, the shallowing of seal dives began ≈ 25 to ≈ 60 days before the arrival of advected phytoplankton (depending on subsidy source: Fig 3, lower right panel).

Similarly, lines 217-221 of results describe temporal and spatial variability in the data, and the extent to which it might support a correlation between the shallowing of seal dives and the arrival of advected phytoplankton, and states that “depth was better explained by the date of advected phytoplankton arrival than calendar date [except for Northeast]” which, though true in relative sense, it detracts from the stronger evidence for resource pulse arrivals lagging behind seal shallowing (lower right panel of Fig 3, Fig 4 and the lower panel of Fig 6). Why not just streamline the text to highlight those lags, as current lines 224-227 do?

These issues carry on to the discussion, which states (lines 247-249) that “The inter- and intra-annual seal shallowing only roughly coincided with estimates of advected phytoplankton from the stations, suggesting that the phenology of these key processes is a complex issue.” This ambiguous statement should be deleted altogether, and instead the section should lead with the key conclusion from lines 261-262: “there is a delay between seal shallowing and the resource pulse that cannot be explained by our existing understanding of the marine ecosystem.” This lag is an important finding. As the discussion already examines, the lag may be influenced by oceanographic variables that either are not being accounted for or poorly represented. Perhaps it is also worth exploring hypothesis derived from a game theoretic perspective (e.g. reference 1), in which the prey of seals might be avoiding peak resource aggregations to optimize risk-energy tradeoffs, which in turn might contribute to an apparent mismatch between seal shallowing and the arrival of advected phytoplankton.

Perhaps my concern is epitomized by Figure 6. The lower panel shows data that clearly indicate that seal shallowing begins and ends before the arrival of advected phytoplankton, yet those shallow dives are lined up with the portion of the upper panel that represents the expected duration of the resource pulse. Yet expectation and data did not match, and this figure muddles that. Having said that, the intended concept of the figure is great; it should be easy to redraw it for the upper panel to be consistent with the data.

Analyses that disentangle the relative contribution of ice break-out from that of advection of subsidies are a strength of the paper. But I also find that the writing should improve for that part of the narrative. For instance, in their response letter to the earlier reviewer, the authors state that "Advection is faster than break-out, which could explain why local break-out occurs substantially later than seal dive shallowing." This important statement should be worked into the abstract and discussion.

Other comments

Lines 76-78 (Introduction) state that "Behavioral theory suggests that intermediate trophic levels such as zooplankton and fishes will maximize fitness by balancing ecological trade-offs between resource acquisition and predation risk, which both decrease with increasing depth." This perspective on the tradeoffs inherent to depth choice is too narrow; cases in which both risk and potential energy gain increase at greater depth are entirely plausible (e.g. reference 2).

Lines 92-97 (Introduction) state that: "We hypothesized that during spring, when resource stratification was weak, predation risk would control the food chain distribution, and intermediate trophic levels (zooplankton and fish) would have deeper distributions. However, we anticipated that a strong resource pulse at the ocean surface during summer would cause these consumers to utilize shallower habitat because the benefits of consuming the abundant resource would out-weigh the risk of predation from air-breathing predators at the surface." The authors did not test this hypothesis, as they have no data on zooplankton and fish vertical distributions. Rather, their statement is a potential interpretation of results better suited for the discussion.

Lines 124-125 (Methods): "...defined the date of sea ice break-out as the first occurrence of a 7-day running mean ice concentration of <50% at each gridded data location." Specify here the size of grid cells (25x25km?)

Lines 125-129 (Methods): Are the coordinates the center points of grid cells?

Lines 145 (Methods): Is the sample size (n=2,941) the number of seal days recorded? Clarify.

Lines 174-176 (Results): "Yearly variation in the timing of shallow seal diving also appeared to alter the depth distribution of lower trophic levels, because earlier ice break-out and the concomitantly earlier resource pulse led to shallower seal diving." This sentence is conceptually muddled because you show no data on "the depth distribution of lower trophic levels."

Lines 176-177 of results state that "earlier ice break-out and the concomitantly earlier resource pulse led to shallower seal diving" and then references Fig. 1, but that figure has no data on ice break-out or resource pulse. The lower panel of Figure 1 (plot of depth of shallowest dive by date of shallowest dive) is not particularly informative and would more directly address the authors point if its X-axis represented dates related to ice breakout or subsidy arrivals.

Lines 234-237 of the discussion state that "Foraging depth of Weddell seals halved and remained shallow for three weeks in mid-summer, suggesting that zooplankton and fishes had shallowed to take advantage of the new phytoplankton availability, despite the predation risk from air-breathing predators." But seals are deepening their dives before the arrival of phytoplankton subsidies! Therefore, the data cannot support that statement. Should this be reframed as a

hypothesis based on in situ production of phytoplankton? Or is that implausible because “local break-out occurs substantially later than seal dive shallowing”?

Minor comments

Line 208. Apparent typo: “proceeded” vs “preceded”

Lines 237-239. There is a wealth of literature supporting the notion that bottom time (and, potentially, net energy gain) increases with the shallowing of dives. You might consider citing one of those papers here.

Figs 2 and 3. The term “ice” is attached to each location name on the text embedded in the figure (And table in Fig3). This is confusing, as the locations are associated with the timing of both ice break-out and advection. Is the term “ice” really meant to be there?

Fig 3 The map should include labels for Cape Royds, Erebus Bay, Ross Island

Fig 5. Caption should specify that each line in panels B and C represents an individual seal.

References

1. Lima, S. L. Putting predators back into behavioral predator-prey interactions. *Trends Ecol. Evol.* 17, 70-75 (2002).
2. Frid, A., Burns, J., Baker, G. G. & Thorne, R. E. Predicting synergistic effects of resources and predators on foraging decisions by juvenile Steller sea lions. *Oecologia* 158, (2009).

Author's Response to Decision Letter for (RSPB-2020-2817.R0)

See Appendix D.

RSPB-2020-2817.R4 (Revision)

Review form: Reviewer 1 (Alejandro Frid)

Recommendation

Accept with minor revision (please list in comments)

Scientific importance: Is the manuscript an original and important contribution to its field?

Excellent

General interest: Is the paper of sufficient general interest?

Excellent

Quality of the paper: Is the overall quality of the paper suitable?

Good

Is the length of the paper justified?

Yes

Should the paper be seen by a specialist statistical reviewer?

No

Do you have any concerns about statistical analyses in this paper? If so, please specify them explicitly in your report.

No

It is a condition of publication that authors make their supporting data, code and materials available - either as supplementary material or hosted in an external repository. Please rate, if applicable, the supporting data on the following criteria.

Is it accessible?

Yes

Is it clear?

Yes

Is it adequate?

Yes

Do you have any ethical concerns with this paper?

No

Comments to the Author

I thank the authors for their revision, which has satisfactorily addressed my earlier criticisms. There are some issues of presentation that I will leave the copy editor to address. (For instance, it is odd that the sequence in which figures are first introduced to the text are Figs 1, 5, 4, 2, 3 and 6.)

My only suggesting is to revise conclusion line 312 as "In summary, seal diving behavior suggests that ice break-out and resource pulses..."

Congratulations on a nice paper.

Decision letter (RSPB-2020-2817.R1)

19-Feb-2021

Dear Dr Beltran

I am pleased to inform you that your manuscript entitled "Ice break-out, phytoplankton blooms and the vertical foraging behavior of a Southern Ocean top predator" has been accepted for publication in Proceedings B.

Open Access

Paper charges

Sincerely,

Dr Sasha Dall

Associate Editor:

Board Member: 1

Comments to Author:

The only remaining issue raised is to adjust the conclusion line 312 as "In summary, seal diving behavior suggests that ice break-out and resource pulses..."

Appendix A

DEPARTMENT OF ECOLOGY & EVOLUTIONARY BIOLOGY
CENTER FOR OCEAN HEALTH
LONG MARINE LAB
100 SHAFFER ROAD
SANTA CRUZ, CALIFORNIA 95060

11 March 2021

Dear Editorial Board,

We would like to re-submit our manuscript “**Resource pulses shift the vertical distribution of a Southern Ocean food chain**” (RSPB-2020-0377) as a research article in Proceedings of the Royal Society B.

We are grateful for the thoughtful reviews provided by the two referees and the Associate Editor. We have incorporated the requested revisions and responded to each comment in the postscript of this letter, in **bold**.

Yours sincerely,

Roxanne S. Beltran

Associate Editor

- Both reviewers found the study very interesting, well-executed and well-written. A major weakness of the study is, however, that several inferences had to be made that were not directly based on in situ observations. In this context, Reviewer 2 had some major comments particularly about the apparent mismatch in timing between the change in diving depth and the sea-ice break out, hence putative algae bloom. Furthermore, she/he pointed to the need to strengthen parts of the reasoning that are based on indirect information (from other sites). Moreover, this reviewer suggested an alternative hypothesis to explain your data where ocean current velocity is the primary determinant of in situ phytoplankton production timing. This alternative explanation should be explicitly considered. I also agree that terminology such as “bottom-up/top-down control” of the food chains should be avoided given that most of the processes associated with such control have not been explicitly quantified in current study system.
 - **We appreciate your thoughtful evaluation of our work and have revised the manuscript to address these concerns, as detailed below.**

Reviewer 1

- This is a well-written and nicely documented paper. Well done. I've no major concerns and my only a handful of minor comments, so let's get to them.
 - **We thank you for your comments.**
- It would be helpful to ensure all figures, including supplemental figures, are cited and cited in the order they are discussed in the text. For example - Fig 5a is referred to

indirectly only after 5b and 5c are introduced; Figure S1 is not referenced in the text; Fig 3A is not referenced in the text; Figure S3 is referenced before any other.

- **Thank you for noticing this. We have added references to Figure 5 (L212), Figure S1 (L145) and Figure 3 (L190). We have also re-ordered the supplemental figures and revised all references in the text. All figures (including supplemental figures) are now listed in order.**
- Line 111-112 Are the dates of pupping and molt critical for this analysis? might the end of this sentence be deleted?
 - **Thank you for pointing this out. We have removed the mention of pupping and molting dates, and have moved the reference up to L103 for more details about study animal selection.**
- To help with figure numbering issues noted above and reference Fig 3A (indirectly, at least), I suggest you refer to Figure 3 at the end of the sentence on line 169.
 - **We have made this change, thank you.**
- Line 190-192. Fair point here, but I feel like a more convincing argument could be made if you can link to other ancillary evidence from prior tracking studies in the Ross Sea or elsewhere. Do those studies support your assumption that the seals remain reasonably stationary during the period of your tracking.
 - **Our recent paper (Beltran et al. 2019 Scientific Reports) includes multiple repeat sightings of these individuals throughout Jan/Feb which provides supporting evidence that seals remain stationary during this period. We have added this citation to the manuscript.**
- Line 201 - it may be worth clarifying that foraging efficiency increases because more time can be spent foraging relative to ascent and descent on shallow dives. Presumably total dive time or average dive time did not change much during across the tracking period?
 - **Thank you for pointing this out. We have amended the sentence to read “Diving efficiency (the proportion of time spent in the bottom phase of each dive) nearly doubled during this period because shallower dives require less descending and ascending transit.**

Reviewer 2

- This MS investigates inter-annual differences in the progression of the dive foraging pattern of Weddell Seals and relates these to the timing of ice breakout as a proxy for phytoplankton bloom timing. It also shows, via stable isotope analysis, that the diet of these seals remains relatively constant despite changes in dive depth. This is presented as evidence of a seasonal vertical restructuring of the foodweb in response to the bottom-up effect of the “resource pulse” delivered by the phytoplankton bloom. I found the paper mainly well written with a clear narrative. I have no concerns about the methodology. It is a perfectly good study of the relationship between diving behaviour and ice-dynamics, which is enhanced by stable isotope data showing that diet remains relatively constant. This supports the interpretation that predator dive distribution tracks prey vertical distribution. This in turn raises some interesting questions about what drives these changes.
 - **We appreciate your careful evaluation of the paper and this clear summary.**
- Although the authors provide convincing evidence that these changes are associated with the timing of ice break-out, their proposed mechanism does not fit the evidence. I also have some concerns about the ambiguity of language around key information about spatial scale and what constitutes a “resource pulse”.
 - **Thank you for your comments. We have made several changes in response to these points below and feel that they have increased the quality of the manuscript.**

- The authors have used their observations of predator diving behaviour to infer a sequence of events affecting a whole food chain. I feel that the inference stretches the data beyond its capabilities. The paper would be stronger if it included data to support these inferences, particularly about the phytoplankton bloom. I acknowledge that the MS provides some information (Fig S2) about the relative timing of sea ice and phytoplankton, but this is at a different location and in a different year from the study. Perhaps more clarity about the relationship between these data and the inferences the authors are making would help.
 - **Thank you for your comments. We agree that local phytoplankton data would be preferable but unfortunately those data do not exist. In the revised manuscript we have clarified the relationship between the data on sea ice breakout and phytoplankton from the nearby location, Cape Crozier (Fig S4) and phytoplankton dynamics at our site (see revised figure 3A). We have also softened the language around top-down/bottom-up control based on comments below (e.g., abstract L40 and discussion L225-226 and L281-282). We feel the novel contribution of our dataset is the cascading vertical movement, and we have tried to make that clearer. We hope these revisions alleviate your concerns.**
- Unfortunately though, the main conclusions that the “extreme near-surface resource pulse ... resulted in prey shallowing” [40] and “sea ice break-out appears to trigger cascading vertical migrations... shifting control... to bottom up”[195] [also 206], seem to contradict the evidence of Fig 3 which shows that shallow diving (which is presented as an indicator of prey vertical distribution) begins before ice break-out. This shallow diving is described as occurring “for three weeks during the phytoplankton bloom” [198]. Yet “the ice break-out date was considered the approximate start date of in situ primary production” [135]. If anything, break-out seems to mark the beginning of a return to deeper diving.
 - **Thank you for your comment. As noted above, we have clarified the relationship between sea ice breakout and phytoplankton in Fig S4 and Fig 3A.**
- The ice break-out information is for a location 40km distant from the study site, but there is no discussion of how representative this is of the study site. The data (Fig S2) suggest that peak production follows ice break-out and chlorophyll is lower 5 days before ice break-out than during the post-bloom period two months later (i.e. there is no evidence of a phytoplankton bloom that occurs during the period of shallow diving before ice break-out).
 - **We have clarified the timing of sea ice breakout and phytoplankton in the methods and supplemental material. The reviewer is correct that Figure S4 previously showed peak phytoplankton production occurring after ice breakout. However, this data was from 150km away from our study site. The current velocity far exceeds the rate of ice break-out so advection is likely to carry phytoplankton into our local area far before local ice breakout.**
- Fig 1 shows that minimum dive depth occurs before 1st January and is followed by a rapid return to deeper diving, whereas Fig S6 shows that ice break-out was always after 1st January. The claim that “At the end of the summer, when... phytoplankton biomass was depleted... seals resumed foraging at (deeper) depths” is not supported by this evidence. It looks like seals begin shallow foraging before midsummer and return rapidly to deep diving just as the putative bloom develops.
 - **You are correct. This sentence was based on the assumptions described above (i.e. phytoplankton is advected into the study area before the local in-situ production occurs); however, given the lack of data to confirm this assumption, we have removed it. We have instead put this claim forward as a hypothesis. The sentences now read: “At the end of summer seals resumed foraging at these depths (Figure 6). This deepening may coincide with nutrient limitation (64) and phytoplankton biomass depletion (Figure S2), when mesopelagic fishes (32, 65) and krill (30) are thought to inhabit deeper depths.”**

- The authors do not explicitly acknowledge this mismatch in timing between observations and the expectation based on their assumptions. The authors allude to the possibility of earlier phytoplankton availability ([135-137], Fig S2 caption) but this is not sufficient to resolve the inconsistency. Also the language around primary production/phytoplankton bloom and timing is ambiguous. Are “in situ primary production” and “phytoplankton bloom” synonymous? How is ice break-out an “estimate of primary production” [136], and what does “conservative” mean in this context? Do you mean it is an estimate of the start of the bloom, but is likely to be late? The authors need to be clear and consistent in describing events that are central to their argument.
 - **Thank you for pointing out these points of confusion. Phytoplankton bloom usually refers to a high concentration of phytoplankton in an area regardless of where the production originated (e.g., *in situ* production or production advected from elsewhere). Thus, while sea ice breakout triggers *in situ* production, the actual bloom may precede the date of ice breakout because it is advected from elsewhere. We have clarified this throughout the text (e.g., L153-155) and added the following to the Supplemental Materials: “The current speeds in the local McMurdo Sound region have been estimated at 6.57 kilometers per day (3). Thus, phytoplankton advection from the Crozier area to McMurdo Sound should take approximately 23 days to reach McMurdo Sound, a distance of 150km. After the sea ice breaks out, *in situ* phytoplankton production markedly increases as nutrients are released and ice no longer shades the photic zone (4). Advective inputs are thought to contribute significantly more to water column productivity than *in situ* production. Thus, the vertical distribution shift of zooplankton and fishes proposed here could begin as soon as water column chlorophyll begins to increase, which likely precedes the date of ice break-out (Figure S4).”** The new Figure S4 includes a clearer explanation of these processes and should help explain the mismatch between observations and expectations. In the text, we have replaced “conservative estimated of primary production” with “early estimate for the date of peak primary production”. We have also added Figure 3 panel A so readers can directly compare the seal diving pattern with estimated Chlorophyll concentration in the local area.
- Also, the following section of the Supplement lacks any information which allows the reader to relate the sequence of events to the timing metrics (date and ice break-out date) used in the MS. Rather, it suggests that ocean current velocity is the primary determinant of in situ phytoplankton production timing: “Each summer as sea ice degrades, three sequential processes occur. Initially, sea ice microalgae are released into the water column and can photosynthesize at very low irradiance levels; however, these microalgae contribute little to standing chlorophyll concentration. Next, phytoplankton are advected into the Erebus Bay study area (77.6°S 167.0°E) from the Ross Sea polynya (centered approximately at 77.0°S 175.0°E). About two weeks later (based on ocean current velocity), in situ phytoplankton production markedly increases as nutrients are released and ice no longer shades the photic zone (3).”
 - **We appreciate your comment and realize now that this supplemental text was confusing. The two weeks between step 2 and 3 are calculated based on current velocity data (as described in the supplemental material). We should have explicitly mentioned that ice break-out is associated with the 3rd step and that the 2nd step should occur several weeks *prior* to ice breakout based on the current data. We have amended the supplemental text (see changes above).**
- So, it seems that the data do not support sea ice break-out as the trigger for deep diving (in contrast to the statements on lines 195 and 206), or an “extreme resource pulse” in the form

- of a local phytoplankton bloom as the mechanism (in contrast to the central theme of the MS). Rather the data suggest that both of these events mark the end of shallow diving.
- **We agree that sea ice breakout at our study site coincides with the end of shallow diving. We have clarified that advected phytoplankton reaches our study site much earlier (~32 days) than ice breakout and this phytoplankton likely triggers the beginning of shallow diving. We have added a panel to Figure 3 illustrating the timing of advected phytoplankton to clarify the sequence of events.**
 - The paragraph quoted above suggests that there could be some elevation of phytoplankton at the study site before ice break-out, as a result of algal release elsewhere. It could be the arrival of this phytoplankton which triggers shallow diving. This raises a question about why this arrival triggers the opposite response from the onset of a local bloom.
 - **We agree that it is puzzling that diving deepens as local sea ice breakout occurs. One possible explanation is that local nutrient resources are depleted by the advected phytoplankton resulting in limited *in situ* phytoplankton production. We have raised this as a hypothesis to be tested.**
 - Moreover, the authors present evidence from the literature for other sea ice ecosystems that supports their expectation that phytoplankton blooms are associated with shallower prey distribution. It is therefore interesting to ask why the evidence is not so clear cut in this case. Is it that the chosen metric is a poor indicator of the bloom or is this ecosystem behaving differently?
 - **Thank you for pointing this out. We have clarified above how the coincidence of timing and phytoplankton in our study are similar to other studies.**
 - Thus, I think it would be possible for the authors to develop a hypothesis that better fits the evidence. However they would be well advised to avoid repeating the mistake of focusing on the hypothesis rather than the strengths of their data.
 - **Thank you for this suggestion. We have clarified the temporal patterns as described above to illustrate how dive patterns coincide with advected phytoplankton.**
 - I have two more minor comments about language and interpretation:
 - The metric in Fig 3c is largely a function of dive depth (as travel time is less for shallow dives) and it is inappropriate to describe it as “foraging efficiency”. Use a descriptive term similar the one in parentheses in this figure.
 - **Thank you for pointing this out. We have changed the terminology in this figure and the text from “foraging efficiency” to “diving efficiency” to reflect that shallower dives require less ascending and descending transit time. “Diving efficiency” has been used to describe this metric in other studies (eg., Zimmer 2010 Aquatic Biology, Cornick 2006 JEMBE).**
 - This may be a matter of taste but the language around “bottom-up/top-down control” of the food chains implies a set of controls on trophic relationships and/or population dynamics. Thus it seems inappropriate to use this terminology to describe the changes in vertical distribution being discussed, especially as the authors are arguing that feeding relationships are unaffected by these changes.
 - **We appreciate your comment. We have removed this wording and focus the manuscript on the vertical distributions of predators and their prey.**
 - [37,38] The geographical terms are potentially misleading: “a” rather than “the” Southern Ocean food chain would be more appropriate given the localised scale of the study. Similarly, “the annual phytoplankton bloom in Antarctica” generalises the study to oversimplify the complex dynamics of a vast area. Also “Antarctica” is a continent.
 - **We appreciate these suggestions. We have amended the text to read “...shifts in a Southern Ocean food chain before, during, and after the annual phytoplankton bloom”. We have moved the mention of Antarctica to L42.**

- [62] I suggest “concentrates” instead of “aggregates”; delete “and their offspring” which is redundant.
 - **Thank you for these recommendations. We have changed the text as suggested.**
- [134] Figure S2?
 - **We have added a mention of Figure S2w to L139.**
- [207] “trigger”
 - **Thank you for catching our grammatical error. We have changed “triggers” to “trigger”. Please note that this change is now on L215 due to the edits above.**
- Fig 3: This would be improved by the addition of sea ice cover and chlorophyll data (as S2).
 - **We have added a panel to this figure illustrating the timing of phytoplankton advected to the study site.**

Appendix B
DEPARTMENT OF ECOLOGY & EVOLUTIONARY BIOLOGY
 CENTER FOR OCEAN HEALTH
 LONG MARINE LAB
 100 SHAFFER ROAD
 SANTA CRUZ, CALIFORNIA 95060

11 March 2021

Dear Editorial Board,

We are pleased to resubmit our revised manuscript “**Resource pulses shift the vertical distribution of a Southern Ocean food chain**” (RSPB-2020-0377 and RSPB-2020-0854) for consideration as a research article in Proceedings of the Royal Society B.

We are grateful for the thoughtful reviews provided by the referees and the Associate Editor. We have incorporated the requested revisions and responded to each comment in the postscript of this letter, in **bold**.

Sincerely,

Roxanne S. Beltran

Associate Editor Board Member:

- While important improvements were made in this resubmission, the paper still needs considerable improvement to reach the level of excellence expected for PRSB. The major comment of the reviewer remains that your key pattern is based on a correlative relationship between ice break-out and phytoplankton availability, which is presented as causal, and moreover still lacks clarity. The reviewer makes excellent suggestions how to rephrase the sequence of events and the differences between drivers and indicators generating the observed patterns. Importantly, to make your story more convincing the suggested analysis to test the hypothesis that advective arrival of material is more important than local *in situ* production will be pivotal to improve your study.
 - **We thank you for your thoughtful evaluation of our work. Below we describe the major revisions that we have made to the text and figures. We feel that these changes have greatly clarified the following issues:**
 - **The correlative, not causative, relationship between ice break-out and phytoplankton availability**
 - **The sequence of events that defines the “resource pulse” along with more consistent terminology throughout the manuscript and figures (e.g., data are now shown in relation to the resource pulse rather than local ice break-out).**
 - **Evidence that the advective arrival of material is more important than local *in situ* production, from both our work as well as that of Rivkin (1).**
 - **We hope that these changes satisfy the remaining concerns about our work. Again, we appreciate your consideration and look forward to hearing your decision on this manuscript.**

Reviewer 2:

- Thank you to the authors for taking some of my comments on board. The manuscript has improved and the revised figure 3 helps to clarify the relationships between seal behaviour, the timing of ice break-out at McMurdo sound and putative phytoplankton availability (but the main text needs to provide a clear and referenced justification for using this temporally shifted phytoplankton data from elsewhere – see comments on para beginning line 127 and Fig 3). As in my previous comments, a critical feature of these relationships is that ice break-out occurs AFTER peak phytoplankton availability. Unfortunately the paper is still confused about this point and it argues against logic that ice break-out is an “early estimate for the date of peak primary production” (line 131) and that it “drives” the observed behaviour (line 150). To be frank, the paper is built round a correlative relationship which is presented as causal but which clearly can not be causal.
 - **We appreciate your thorough evaluation of our work and feel that your suggestions have significantly improved the manuscript. Below we provide details of our revisions, which we hope have eliminated your concerns. To summarize, we have provided a clear and referenced justification of using the temporally shifted phytoplankton data in the main text, and have revised all figures and text to provide the seal diving data in relation to the resource pulse rather than local sea ice break-out. We believe that this revised manuscript clearly describes the relationship as correlative rather than causal.**
- There is a coherent explanation of the results which can be constructed from the information provided (see comments on para beginning line 127) as follows: the TIMING OF ICE BREAK-OUT at McMurdo Sound is an indicator of large scale processes that drive the timing of seasonal events across this part of the Southern Ocean. These include events which occur earlier in the season such as the timing of the phytoplankton bloom at Cape Crozier. It is likely that ADVECTIVE ARRIVAL OF MATERIAL from this bloom at the study site causes a resource pulse at the study site which drives changes in the distribution of prey and hence the foraging behaviour of Weddell Seals. While IN SITU PRODUCTION at the study site might increase after ice break-out, it seems insufficient to prevent the prey from returning to deeper waters.
 - **This is exactly right. We have added this text to both the results and discussion of the manuscript (e.g., L126, L133, L207).**
- The narrative offered by the authors blurs the boundaries between the three processes highlighted in upper case above and uses the phrases “phytoplankton boom” and “primary production” to describe any or all of them. In fact the new edits include changing the phrase “ice break-out” to “phytoplankton bloom” (line 153) even though the authors argue “the actual bloom may precede the date of ice breakout” (response to reviews) and, in the same paragraph (and line 131), that ice break-out is an “early estimate for the date of peak primary production”. So the authors seem to be claiming that ice break-out is simultaneously equivalent to, after, and before the resource pulse! To improve the manuscript based on the current analysis, the authors must:
 - (1) Recognise that local ice break-out may be an INDICATOR of timing but can not be a DRIVER of events that precede it.
 - (2) Be clear in the main text about the reasoning linking their data (TIMING OF ICE BREAK-OUT) to the processes it putatively indicates (ADVECTIVE ARRIVAL OF MATERIAL, IN SITU PRODUCTION) and the evidence that supports this.
 - (3) Use language that clearly distinguishes between the three processes. I concede the authors’ point that “Phytoplankton bloom usually refers to a high concentration of phytoplankton in an area regardless of where the production originated” but I find “resource pulse” a much better term to describe the combined effect of ADVECTIVE ARRIVAL OF MATERIAL and any co-occurring IN SITU PRODUCTION. My advice is to use one phrase consistently and to define it at first use. ADVECTIVE ARRIVAL OF MATERIAL should not be described as (peak) (primary) production since the production happened in the past and elsewhere.
 - (4) Be clear (in the paper itself rather than the supplement) about the spatial relationship between their study site, the location of the ICE BREAK-OUT data and the location where the advected material originates.
 - **We agree that resource pulse is a better term, and have used this throughout the rest of the paper and in the revised figures. We have also re-written this paragraph (L126-147), which now reads “Because direct measurements of**

phytoplankton bloom timing were not available at our study site, we used ice break-out timing to estimate resource pulse timing using the following evidence. We obtained satellite-derived sea ice concentration (US National Snow and Ice Data Center; NASA Bootstrap Sea Ice Concentrations from Nimbus-7 SMMR and DMSP SSM/I-SSMIS, Version 3) and defined the date of sea ice break-out as the first occurrence of a 7-day running mean ice concentration of <50% in the local McMurdo Sound area (2) (Figure S3). The timing of ice break-out at the McMurdo Sound study site is an indicator of large-scale processes that drive the timing of seasonal events across this part of the Southern Ocean. As sea ice extent decreases, phytoplankton concentration increases via *in situ* production as nutrients are released and ice no longer shades the photic zone (3). This enhanced *in situ* production following ice break-out typically occurs 54 days earlier at Cape Crozier (approximately at 77.5°S 169.3°E) than the McMurdo Sound study site (77.6°S 167.0°E) (1) (Supplemental Materials). Ocean currents advect phytoplankton from Cape Crozier to the study site (~150 km distance) in approximately 23 days based on current speeds of 6.57 kilometers per day measured in the local McMurdo Sound region (4). Thus, increasing chlorophyll in the McMurdo Sound study area is likely to occur about 31 days before local ice break-out (5) (Figure S4). The ice-break out then supplements this resource pulse by catalyzing a marked increase in *in situ* production at the study site; however, advective inputs are thought to contribute significantly more to water column productivity than *in situ* production (1). Hereafter in the text and figures, we use the term “resource pulse” to describe the combined effect of the advective arrival of phytoplankton from the Cape Crozier phytoplankton bloom and the local *in situ* production, beginning around 31 days before local ice break-out.”.

- With more clarity about the sequence of events and the differences between drivers and indicators the study might be publishable without any further analysis. It would be more complete (and therefore better suited to a high impact journal) if the authors were able to test the hypothesis that ADVECTIVE ARRIVAL OF MATERIAL is more important than local IN SITU PRODUCTION. This could be done by constructing alternative GAMMs using a direct indicator of bloom timing at Cape Crozier as an explanatory variable. Does this explanatory variable perform better than TIMING OF ICE BREAK-OUT in McMurdo sound? Is the direct indicator strongly correlated with TIMING OF ICE BREAK-OUT in McMurdo sound?
 - To address this concern, we have revised Figure 1 (dashed lines), Figure 2 (x-axis), Figure 3 (x-axis), and Figure 6 (x-axis). We feel that in conjunction with the changes in the text described above and below, the manuscript is now clear on the temporal relationship between the timing of seal shoaling and the resource pulse. More specifically, it is clear that seal shoaling begins just after the resource pulse and continues for several weeks. Unfortunately, the GAMM variable comparison is beyond this study's scope because the data are not available.
- As part of the review process I was asked to check data availability at <http://www.usap-dc.org/view/dataset/601137>. The data fields available are Days since 1 January, Daily mean dive depth and Season. The data required to plot Figs 3C, 3D, 4 and 5 do not seem to be available.
 - Thank you for pointing out our inadvertent omission. We have submitted two additional spreadsheets (and added several rows to the existing metadata) so that all figures can now be reproduced. We added links for both datasets in the data availability section.
- 38 – “deep waters” is misleading; “at depths > X_m” would be better, or “at depth” might work.
 - We have changed “deep waters” to “at depth”.
- 59 - I suggest “in” rather than “along” the marginal ice zone.

- **We have made this change.**
- 64-65 – I suggest continuing the penultimate sentence with “and it is therefore important to understand how ecological dynamics vary across time and 3-dimensional space.”, and ending the paragraph there. There is no intervention that will fix (address) this specific issue.
 - **Thank you for the suggestion. L63-65 now read “A major concern is whether climate change will alter species interactions to create temporal or spatial mismatches that compromise fitness (6, 7). It is therefore important to understand how ecological dynamics vary across time and 3-dimensional space.”**
- 68 – replace “depth-driven clines” with “depth gradients” or similar. This is to remove the debatable assertion that depth is the main driver of all of these variables.
 - **Thank you for pointing this out. The sentence now reads “... with depth gradients of temperature, light, nutrients, and oxygen...”**
- 74 – add “increasing” before “depth”.
 - **We have made this change.**
- 78 – I suggest deleting the following: “which allows them to capitalize on the proximity of their food and oxygen supplies and thus optimize foraging efficiency.” This is an unnecessary level of subjective interpretation of the fact that predators forage where there are prey.
 - **The sentence now reads “These diel vertical migrations (8) are tracked by air-breathing vertebrates (9-11); however, it is unknown whether ephemeral resource pulses cause analogous cascading migrations on a seasonal timescale (12).”**
- 82 – I suggest “use of vertical space”
 - **Thank you for this suggestion. We have changed the text.**
- 83 – The phrase “the Southern Ocean’s most productive region” implies a relevance which isn’t demonstrated. The Ross Sea sector may be the Southern Ocean’s “most productive region” based on averages over millions of square km but that doesn’t mean that your Weddell seal foraging site is more productive than sites in other sectors. Either delete the phrase or rephrase to clarify the difference in scales, along the lines, “at a study site in the Ross Sea, which is in the most productive sector of the Southern Ocean”
 - **We have revised the sentence based on this comment and the preceding. The sentence now reads “Our aim was to understand how resource pulses influence top predators’ use of vertical space at a study site in the Ross Sea, which is in the most productive sector of the Southern Ocean (3, 13).”**
- 127 – This paragraph is still not providing all of the information required to understand ice break-out date as a proxy for resource pulse timing at the study site. Please make sure it contains all of the following info (gleaned mainly from your supplementary information): (1) You do not have direct observations of phytoplankton availability at the study site; (2) “As sea ice extent decreases phytoplankton concentration increases via in situ production”; (3) “After the sea ice breaks out, in situ phytoplankton production markedly increases as nutrients are released and ice no longer shades the photic zone”; (4) This (enhanced in situ production following break out) typically occurs 54 days earlier at Cape Crozier than the study site, the resultant phytoplankton is advected to the study site in approximately 23 days and therefore the “resource pulse” at the study site is likely to start about 31 days before ice break-out; (5) Ice-break out supplements this resource pulse by catalysing a “marked increase” in in situ production at the study site; (6) but “Advective inputs are thought to contribute significantly more to water column productivity than in situ production”. All of these assertions require supporting references.
 - **Thank you for writing this out. We have re-written this entire paragraph per your suggestions and have moved text and references from the supplemental materials. Please see comment above for full text of paragraph.**
- 131 – “The ice break-out date is an early estimate for the date of peak primary production”. Surely it is a late estimate. Your fig 3 implies the peak is between 30 and 10 days BEFORE ice break-out.
 - **We apologize for this oversight. Our intention was to say that ice break-out data is an early estimate for *in situ* primary production. Per the previous revision, this entire paragraph has now been replaced and the sentence no longer appears.**

- 150-158 – This paragraph is about general patterns and does not mention differences between years, so it should refer to Fig 3 rather than Fig 1. This would also allow the authors to put specific dates relative to ice break-out on the various statements.
 - **Thank you for pointing this out. We have referred to Figure 3 rather than Figure 1 in this statement. Please note that this change required us to switch Figures 2 and 3 so that the figures would be referenced in order throughout the text. Thus, this particular sentence now refers to Figure 2.**
- 153 – The phrase “phytoplankton bloom” is used here to replace “ice break-out” (in the earlier version of the MS). I prefer the latter which is the tangible variable analysed, as opposed to the former which is difficult to pin-point in time, but which doesn’t seem to be coincident with ice break-out. However I suggest deleting this whole sentence and replacing it with a statement about the period ending about 31 days before ice break-out which seems to be a reasonable definition of before the bloom.
 - **We have made this correction based on this suggestion and the suggestions above. The sentence now reads “Mean dive depth before the resource pulse was 233 m with only 39% of dives being < 200 m, suggesting that prey species were found at depth.”**
- 160-165 – Again I don’t support the use of the phrase “phytoplankton bloom” as a synonym for “ice break-out”, please revert to the original.
 - **Please see above changes. This now refers to “resource pulse” which matches the text and figure x-axis. Please also note that we are not using “resource pulse” as a synonym for “ice break-out” here – we have simply changed the sentences to describe the seal patterns in relation to the resource pulse rather than ice break-out.**
- Fig 3 – this needs a clear statement that the chlorophyll data are from Cape Crozier but are shifted by 23 days on the x-axis to represent transport time to the study site. The justification for this, including supporting references, should be in the main text. The figure could be improved further with a representation of in-situ production at the study site (the text suggests three key issues to capture: In situ production starts before break-out, it increases at break-out, but it contributes less to the resource pulse than advected material). Given the lack of data, this would have to be diagrammatic.
 - **Thank you for pointing this out. (Please note that due to above changes this revision now refers to Figure 2). We have changed the Fig2A legend to read “Chlorophyll concentration measured at Cape Crozier and shifted by 23 days on the x-axis to represent advection time to the study site in McMurdo Sound (see Supplemental Material)”. Additionally, we have moved a large portion of text (and references) from the supplemental materials to the main text. We feel that this provides more justification and clarification. Finally, we have chosen not to include a conceptual line representing the *in situ* production because due to the lack of data, we do not feel comfortable guessing the concentration. We hope this will suffice.**
- Fig S4. This is useful. The msap could also locate the study site and location of the ice break out data. I suggest labeling Ross Island. Remove the word “dashed” as the lines seem solid.
 - **Thank you for these comments. We labeled Ross Island and added a dot to represent the study site. We have also removed both instances of the word “dashed” to refer to the light and dark green Chlorophyll lines. The locations of the ice break-out data are represented by text.**

References Cited

1. Rivkin RB. Seasonal patterns of planktonic production in McMurdo Sound, Antarctica. *American Zoologist*. 1991;31(1):5-16.
2. Fauchald P, Tarroux A, Tveraa T, Cherel Y, Ropert-Coudert Y, Kato A, et al. Spring phenology shapes the spatial foraging behavior of Antarctic petrels. *Marine Ecology Progress Series*. 2017;568:203-15.
3. Smith Jr WO, Ainley DG, Arrigo KR, Dinniman MS. The oceanography and ecology of the Ross Sea. *Annual Review of Marine Science*. 2014;6:469-87.
4. Barry J, Dayton P. Current patterns in McMurdo Sound, Antarctica and their relationship to local biotic communities. *Polar Biology*. 1988;8(5):367-76.
5. Ackley S. McMurdo Sound, Antarctica: An opportunity for long-term investigation of a high-latitude coastal ecosystem. Workshop Report. San Jose, California; 2004 13-15 April 2004.
6. Inouye DW, Barr B, Armitage KB, Inouye BD. Climate change is affecting altitudinal migrants and hibernating species. *Proceedings of the National Academy of Sciences*. 2000;97(4):1630-3.
7. Kerby J, Post E. Capital and income breeding traits differentiate trophic match–mismatch dynamics in large herbivores. *Philosophical Transactions of the Royal Society B: Biological Sciences*. 2013;368(1624):20120484.
8. Alonzo SH, Mangel M. Survival strategies and growth of krill: avoiding predators in space and time. *Marine Ecology Progress Series*. 2001;209:203-17.
9. Croxall J, Everson I, Kooyman G, Ricketts C, Davis R. Fur seal diving behaviour in relation to vertical distribution of krill. *The Journal of animal ecology*. 1985;54(1):1-8.
10. Kooyman G, Cherel Y, Maho YL, Croxall J, Thorson P, Ridoux V, et al. Diving behavior and energetics during foraging cycles in king penguins. *Ecological Monographs*. 1992;62(1):143-63.
11. Bollens SM, Rollwagen-Bollens G, Quenette JA, Bochdansky AB. Cascading migrations and implications for vertical fluxes in pelagic ecosystems. *Journal of Plankton Research*. 2010;33(3):349-55.
12. Massom RA, Stammerjohn SE. Antarctic sea ice change and variability—physical and ecological implications. *Polar Science*. 2010;4(2):149-86.
13. Arrigo KR, Worthen D, Schnell A, Lizotte MP. Primary production in Southern Ocean waters. *Journal of Geophysical Research: Oceans*. 1998;103(C8):15587-600.

Appendix C

BERKELEY • DAVIS • IRVINE • LOS ANGELES • RIVERSIDE • SAN DIEGO • SAN FRANCISCO

SANTA BARBARA • SANTA CRUZ

DEPARTMENT OF ECOLOGY & EVOLUTIONARY BIOLOGY
CENTER FOR OCEAN HEALTH
LONG MARINE LAB
100 SHAFFER ROAD
SANTA CRUZ, CALIFORNIA 95060

11 March 2021

Dear Editorial Board,

We are pleased to resubmit our revised manuscript “**Resource pulses shift the vertical distribution of a Southern Ocean food chain**” (RSPB-2020-0854.R1) for consideration as a research article in Proceedings of the Royal Society B. Please note that per the reviewer suggestions below, we have changed the title to “Ice break-out, phytoplankton blooms and the vertical distribution of a Southern Ocean food chain”.

We appreciate your patience as we carefully weighed the reviewer’s comments and suggestions. The reviewer made several requests for clarity and additional analyses - now included in the manuscript and supplemental material – which we believe have significantly strengthened the manuscript. Below we have responded to the reviewer’s comments in bold. Due to the extensive nature of the comments and the large number of changes to the manuscript and figures, we have not provided line-by-line revisions. Instead, we have summarized our revision approach.

Sincerely,

Roxanne S. Beltran

I have reviewed earlier versions of this MS. I advised in both reviews that (1) it is essential to be clear and consistent about central concepts in the narrative, including the timing and duration of the resource pulse that putatively drives seal behaviour; and (2) it is not scientifically valid to present ice break-out at the study site as a driver of events that precede it.

We appreciate the reviewer’s careful evaluation of these issues. Following their suggestion, we have extracted more sea ice and current velocity data to carefully characterize the timing and duration of the resource pulse. Details of these analyses are provided below and in the manuscript and supplement. We have also softened the language to reflect the correlative relationship between ice/phytoplankton and seal behavior.

The Methods of the current MS define the “resource pulse” as “the combined effect of the advective arrival of phytoplankton from the Cape Crozier phytoplankton bloom and the local in situ production” (lines 164 to 166), i.e. a protracted process starting approx. 31 days before, and ending some time after, ice break-out. Yet in the Results and figures it is treated as a specific date (i.e. a process lasting no longer than a day) (e.g. lines 183-201), fig caption 3, figs 2, 3, 6). This leads to

particular confusion with lines 187 to 191. The first statement (187) suggests that shallowing is coincident with advection AND in situ production while the second (190) suggests that this PRECEDES a return to deeper diving. Yet the return begins before the event (ice break-out) (S2) which catlayzes “a marked increase in in situ production at the study site” (line 162).

In hindsight, we should have used a rectangle rather than a line to denote the resource pulse. Part of our hesitation to do so was the uncertainty surrounding the date of phytoplankton production and arrival to the local area as well as the phytoplankton bloom’s duration. As noted above, we have re-analyzed the data with respect to all possible combinations of advection rates and ice dynamics to underscore the uncertainty surrounding these processes and the need for more research. We have also included these considerations in the discussion.

The abstract suggests that ice break-out is a driver of behaviour “with later ice break-out delaying periods of seal dive shallowing by one month.” (line 45). I further suggested that “It would be more complete (and therefore better suited to a high impact journal) if the authors were able to test the hypothesis that ADVECTIVE ARRIVAL OF MATERIAL is more important than local IN SITU PRODUCTION. This could be done by constructing alternative GAMMs using a direct indicator of bloom timing at Cape Crozier as an explanatory variable.” The authors have not done this, claiming that “the data are not available”. To clarify, my suggestion was to use ice break-out timing at Cape Crozier as an indicator of bloom timing at that site. I find it surprising that NASA Bootstrap Sea Ice Concentrations are not available for Cape Crozier but are available for McMurdo. Given that the current revision falls short on these three critical points, I do not think it is ready for publication.

We apologize for the miscommunication regarding the request for an additional analysis. We were under the impression that the reviewer was requesting phytoplankton data, which unfortunately do not exist. We re-downloaded the NetCDF files using Python and analyzed ice data from a set of locations that appear to be representative of the hypothesized ice/phytoplankton dynamics. There are many gridded (25x25km) data locations within the vicinity of the Erebus Bay study area (black square in figure below) and Ross Island (center). We chose 5 data locations along the approximate path of ice break-out and phytoplankton advection. Though there is a consensus in the research community that Erebus Bay phytoplankton production likely originates from Cape Crozier and is advected by the Cape Royds current (1), this is not certain. Specifically, sea ice break-out occurs from northeast to southwest (green → grey → blue → yellow → red) and *in situ* phytoplankton production is expected to do the same.

Please note that while data location 71271 appears to be an ideal location given its proximity to the seal colony, it is located too close to the ice shelf (thick black line above) and thus sea ice concentration does not drop below 50% each year. Therefore, we did not include this data location in our analysis.

We calculated the approximate number of days it would take for phytoplankton to be advected to the study area from each of five data locations to the local Erebus Bay study area based on the distance along the coastline. This advection duration (black line in plot below), longer for data locations farther from Erebus Bay, was then added to the mean annual ice break-out date at each data location to estimate the date advected phytoplankton would reach Erebus Bay. These data suggest a staggered arrival of resources (originating from various sources) to Erebus Bay.

The date of shallowest seal diving at Erebus Bay, averaged across the four study years, was January 6 (arrow, above) which coincides with a resource pulse resulting from advection from the North and Northeast data locations. Advection is faster than break-out, which could explain why local break-out occurs substantially later than seal dive shallowing. We discuss the potential problems with these calculations (e.g., the current velocities were measured near the local data location and may be extreme under-estimates of the Cape Royds current velocity) in the manuscript discussion. Given the strong inter-annual dynamics in this ecosystem, we present a very detailed version of the above figure that includes a range of current speeds and phytoplankton origination locations for each year of the study in the main manuscript:

Fig. 2. Temporal patterns of ice break-out, phytoplankton advection, and bloom duration for each of five locations using three published current velocities in the four study years. Seal shallow diving (thick black line) overlaid on ice break-out (circles), phytoplankton advection (dashed lines), arrival of phytoplankton advection (triangles), and approximate bloom duration (solid lines) for each ice location (colors) and year (panels). Each location/year combination has three lines representing various estimates of ocean current velocities, from top to bottom: 12 km/day (2), 10.3 km/day (3), and 6.5 km/day (4).

To reiterate: This is a valid and interesting study. The analyses have merit. However the “explanatory variable” (local ice break-out date) is, at best, a proxy, for events that predate it. If local in situ production is highest after ice break out, then in neither causes nor maintains shallow diving. The authors could consider a potentially more relevant explanatory variable (Cape Crozier ice break out). Nonetheless the onus is on them to provide a clear and consistent explanation of their findings which is also clear about the limitations of their data. I do not think that a series of hasty revisions is the best way to achieve this and I strongly advise that the authors make their own efforts to review the MS for clarity and consistency before the next submission to a journal.

One again, we appreciate your patience and attention to detail, which have significantly improved the quality of our manuscript. We believe that with the analyses we have added and text we have revised, the manuscript now strikes a better balance between the limitations and potential implications of our data. For example, while we do not attempt to quantify the relative contributions of each data location, we do make an educated guess about the likely origination of the phytoplankton that may be driving this seal shallowing pattern. We look forward to the subsequent research projects that may occur as a result of the questions raised from these analyses. We hope you agree that these careful revisions have added consistency and clarity to the manuscript. Thank you for your time.

1. Knox G. Primary production and consumption in McMurdo Sound, Antarctica. *Antarctic Ecosystems*: Springer; 1990. p. 115-28.
2. Palmisano AC, SooHoo JB, SooHoo SL, Kottmeier ST, Craft LL, Sullivan CW. Photoadaptation in *Phaeocystis pouchetii* advected beneath annual sea ice in McMurdo Sound, Antarctica. *Journal of Plankton Research*. 1986;8(5):891-906.
3. Littlepage JL. Oceanographic investigations in McMurdo sound, Antarctica. *Biology of the Antarctic seas II*. 1965;5:1-37.
4. Gilmour A, MacDonald W, Van der Hoeven F. Winter measurements of sea currents in McMurdo Sound. *New Zealand Journal of Geology and Geophysics*. 1962;5(5):778-89.

Appendix D

DEPARTMENT OF ECOLOGY & EVOLUTIONARY BIOLOGY
CENTER FOR OCEAN HEALTH
LONG MARINE LAB
100 SHAFFER ROAD
SANTA CRUZ, CALIFORNIA 95060

11 March 2021

Dear Editorial Board,

We are pleased to resubmit our revised manuscript (RSPB-2020-0854) for consideration as a research article in Proceedings of the Royal Society B. Please note that per the Associate Editor's comments, we have changed the title to: **“Ice break-out, phytoplankton blooms and the vertical foraging behavior of a Southern Ocean top predator”**.

We appreciate your patience as we carefully weighed the reviewer's comments and suggestions. The reviewers made several requests for clarity which we believe have significantly strengthened the manuscript. Below we have responded to each comment in **bold**.

Sincerely,

Roxanne S. Beltran

Associate Editor Board Member

Comments to Author:

The authors considerably improved their revised paper. Nevertheless, I agree with the new Reviewer 2 that still too much ambiguity remains about support for the main conclusion of a direct coupling between ice-breakout, the arrival of advected phytoplankton and the shallowing of diving behavior of Weddell seals. A temporal decoupling of these phenomena precludes making strong conclusions thereby repeating a major comment already made during the first review round. A further, careful re-writing will be needed to avoid over-interpretation and ambiguity which will be crucial to make this paper acceptable. I also want to note that the title gives the wrong impression that data were collected on the entire food chain, which is overselling the study.

Thank you for your helpful comments. We have removed conclusions about coupling between advected phytoplankton and seal shallowing. We have also carefully addressed each point below and feel that the manuscript now tells a clearer story about the vertical behavior of a top predator species in relation to what we know about physical and biological oceanography during the polar summer. Per your suggestion, we have changed the manuscript title to: “Ice

break-out, phytoplankton blooms and the vertical foraging behavior of a Southern Ocean top predator”.

Reviewer(s)' Comments to Author:

Referee: 1

Comments to the Author(s).

Dear Authors,

I reviewed an earlier version of this manuscript, which was largely acceptable to me at that time. Nonetheless, it has improved in the interim and now offers a more nuanced and cautious view of the role of ice-breakout and subsequent phytoplankton blooms on diving behavior of Weddell seals. In general, I think you have done a decent job adapting your analyses to the requests of reviewers and I see the current manuscript as acceptable for publication, noting a handful a few minor suggestions for you to consider:

Thank you for your thoughtful comments.

Line 208 – should be “proceeded” or “progressed”

We have replaced “preceded” with “progressed”.

Line 261 – for the sake of clarity, please specify which characteristic of the ice station most closely matches seal shallowing.

Thank you for pointing out the lack of clarity. We have added the word “temporal”. The sentence now reads: “We found that inter-annual temporal variability in seal shallowing most closely matched that of ice stations closer to the seal colony...”

Lines 217-229 - for each specific result mentioned, it would be very helpful to reference the figure (and panel) that presents the data/result. For example, the first sentence should reference figure 2. On a related note, the map presented in figure 3 might be better housed if combined as a second panel for figure 2 to show the spatial and temporal elements of resource pulsing in one coherent figure.

Great point about moving the map from Figure 3 to Figure 2. We have done that and revised the figure legends to match. We have also added references to figures and panels after every sentence. The revised paragraph is now Lines 416-433.

Line 219-222: While this may be a true statement according to the table in figure 3, it would be more helpful to highlight results for covariates that best explain the pattern in the data, rather than focus on one that doesn't (or, why would we expect Julian day to be a good predictor of a behavior so intimately tied to natural variability in the environment?)

Thank you for pointing out that our explanation of the covariates (i.e., the two most explanatory locations) was buried in that paragraph. We have moved that sentence up so that the explanatory locations are explained first and the weaker covariate (calendar date) is presented down below. These changes appear on L220 to L230.

Referee: 3

Comments to the Author(s).

This interesting paper integrates measures of seal foraging behaviour with oceanographic data to test hypotheses about how the arrival of resource subsidies (advected phytoplankton) cause the vertical redistribution in the water column of multiple trophic levels. The vertical movement of only one trophic level, seals, is measured directly, while that of fish (the seals' prey) and of zooplankton (the fish's prey) is assumed based on the literature. This revision has benefitted from a prior review. I find that the manuscript contains rigorous analyses that should be of general interest to ecologists, oceanographers, and others interested in the ecological implications of global environmental change. However, several statements and inferences are unclear or not well supported, which made me—an interested reader sympathetic to the many complexities inherent to field data and analyses—struggle to discern the unifying themes and main points of the study. This issue can be solved with careful re-writing.

We appreciate your kind and extensive comments and have made our best effort to re-write the manuscript accordingly, with careful attention to your requests. Specific changes are listed below.

My view is that the manuscript would greatly improve if the authors would actively accept, from the outset, that support is weak to inconsistent for the hypothesis that shallowing is triggered by the arrival of advected phytoplankton. The authors rigorously tested that hypothesis, which is no small feat, yet the data lead to the conclusion that (lines 261-262) “there is a delay between seal shallowing and the resource pulse that cannot be explained by our existing understanding of the marine ecosystem.” A revision should highlight evidence for that delay, rather than muddling the narrative by struggling to find evidence for synchrony between seal behaviour and the arrival of subsidies, as the current version does. For instance, the abstract, states that “dive shallowing often coincided with annual ice break-out and advection of the phytoplankton bloom from the surrounding areas,” but this is ambiguous. Instead, the abstract might highlight that, averaged across all years, the shallowing of seal dives began ≈ 25 to ≈ 60 days before the arrival of advected phytoplankton (depending on subsidy source: Fig 3, lower right panel).

Thank you for this suggestion – we agree that this is the best approach. We have made several changes throughout the manuscript. We revised the abstract and the paragraph in question to reflect the mismatch in timing between advection and seal shallowing reflected in Figure 2.

Similarly, lines 217-221 of results describe temporal and spatial variability in the data, and the extent to which it might support a correlation between the shallowing of seal dives and the arrival of advected phytoplankton, and states that “depth was better explained by the date of advected phytoplankton arrival than calendar date [except for Northeast]” which, though true in relative sense, it detracts from the stronger evidence for resource pulse arrivals lagging behind seal shallowing (lower right panel of Fig 3, Fig 4 and the lower panel of Fig 6). Why not just streamline the text to highlight those lags, as current lines 224-227 do?

Please see our previous response; we have re-organized and re-written the paragraph to focus more on the inter-annual variation in timing than on which locations are better for explaining the seal shallowing patterns.

These issues carry on to the discussion, which states (lines 247-249) that “The inter- and intra-annual seal shallowing only roughly coincided with estimates of advected phytoplankton from the stations, suggesting that the phenology of these key processes is a complex issue.” This ambiguous statement should be deleted altogether, and instead the section should lead with the key conclusion from lines 261-262: “there is a delay between seal shallowing and the resource pulse that cannot be explained by our existing understanding of the marine ecosystem.” This lag is an important finding. As the discussion already examines, the lag may be influenced by oceanographic variables that either are

not being accounted for or poorly represented. Perhaps it is also worth exploring hypothesis derived from a game theoretic perspective (e.g. reference 1), in which the prey of seals might be avoiding peak resource aggregations to optimize risk-energy tradeoffs, which in turn might contribute to an apparent mismatch between seal shallowing and the arrival of advected phytoplankton.

Thank you for this comment. We have deleted the ambiguous sentence, and instead focused on the delay by writing the following paragraph (see comments above).

Perhaps my concern is epitomized by Figure 6. The lower panel shows data that clearly indicate that seal shallowing begins and ends before the arrival of advected phytoplankton, yet those shallow dives are lined up with the portion of the upper panel that represents the expected duration of the resource pulse. Yet expectation and data did not match, and this figure muddles that. Having said that, the intended concept of the figure is great; it should be easy to redraw it for the upper panel to be consistent with the data.

Thank you bringing our attention to this inconsistency. We have redrawn the upper panel (the conceptual model) to match the data figures in the manuscript.

Analyses that disentangle the relative contribution of ice break-out from that of advection of subsidies are a strength of the paper. But I also find that the writing should improve for that part of the narrative. For instance, in their response letter to the earlier reviewer, the authors state that “Advection is faster than break-out, which could explain why local break-out occurs substantially later than seal dive shallowing.” This important statement should be worked into the abstract and discussion.

As described above, we have completely re-written this paragraph. We have also added a sentence to the abstract about how advection rates are faster than ice break-out.

Other comments

Lines 76-78 (Introduction) state that “Behavioral theory suggests that intermediate trophic levels such as zooplankton and fishes will maximize fitness by balancing ecological trade-offs between resource acquisition and predation risk, which both decrease with increasing depth.” This perspective on the tradeoffs inherent to depth choice is too narrow; cases in which both risk and potential energy gain increase at greater depth are entirely plausible (e.g. reference 2).

This is a great point. We have added that citation and changed “which both decrease with increasing depth” to “which both vary with depth”. We believe this still maintains the original purpose, which was to demonstrate that lower trophic levels balance ecological trade-offs across vertical gradients.

Lines 92-97 (Introduction) state that: “We hypothesized that during spring, when resource stratification was weak, predation risk would control the food chain distribution, and intermediate trophic levels (zooplankton and fish) would have deeper distributions. However, we anticipated that a strong resource pulse at the ocean surface during summer would cause these consumers to utilize shallower habitat because the benefits of consuming the abundant resource would out-weigh the risk of predation from air-breathing predators at the surface.” The authors did not test this hypothesis, as they have no data on zooplankton and fish vertical distributions. Rather, their statement is a potential interpretation of results better suited for the discussion.

Per suggestions from Reviewer 1, we have re-written this paragraph to remove the inferential text about the reason behind lower trophic level distributions and instead focus on how we hypothesized that changes in prey distributions might influence top predators.

Lines 124-125 (Methods): "...defined the date of sea ice break-out as the first occurrence of a 7-day running mean ice concentration of <50% at each gridded data location." Specify here the size of grid cells (25x25km?)

We specify the size of the grid cells in the sentence prior: "To do this, we obtained satellite-derived daily sea ice concentration (% cover; US National Snow and Ice Data Center; NASA Bootstrap Sea Ice Concentrations from Nimbus-7 SMMR and DMSP SSM/I-SSMIS, Version 3; spatial resolution 25x25km) and defined the date of sea ice break-out as the first occurrence of a 7-day running mean ice concentration of <50% at each gridded data location (43) (Figure S1) which we labeled "Northeast" (centered at 77.24°S 169.10°E), "North" (77.20°S 168.09°E), "Northwest" (77.10°S 166.09°E), "West" (77.32°S 165.84°E), and "Southwest" (77.54°S 165.58°E)."

Lines 125-129 (Methods): Are the coordinates the center points of grid cells?

Thank you for pointing out that omission. We have added the word "centered at" before specifying the latitude and longitude of each location (see above).

Lines 145 (Methods): Is the sample size (n=2,941) the number of seal days recorded? Clarify.

You are correct. We have added "seal-days" after "n=2,941" to clarify.

Lines 174-176 (Results): "Yearly variation in the timing of shallow seal diving also appeared to alter the depth distribution of lower trophic levels, because earlier ice break-out and the concomitantly earlier resource pulse led to shallower seal diving." This sentence is conceptually muddled because you show no data on "the depth distribution of lower trophic levels."

Thank you for the suggestion. We have removed that sentence and replaced it with: "Across years, seal diving was shallower when the date of seal shallowest seal diving was earlier (Figure 1; $Depth_{shallowestdives} = 1.64 (\pm SE = 0.18) * Date_{shallowestdives} + 97.25, n = 4 \text{ years}, R^2=0.95, \text{one-tailed } P=0.015$)."

Lines 176-177 of results state that "earlier ice break-out and the concomitantly earlier resource pulse led to shallower seal diving" and then references Fig. 1, but that figure has no data on ice break-out or resource pulse. The lower panel of Figure 1 (plot of depth of shallowest dive by date of shallowest dive) is not particularly informative and would more directly address the authors point if its X-axis represented dates related to ice breakout or subsidy arrivals.

Please see our response to the last suggestion. The sentence now specifically refers to the content of the figure (depth versus date of shallowest diving), which we feel is interesting from a biological perspective but that we will refrain from attempting to interpret.

Lines 234-237 of the discussion state that "Foraging depth of Weddell seals halved and remained shallow for three weeks in mid-summer, suggesting that zooplankton and fishes had shallowed to take advantage of the new phytoplankton availability, despite the predation risk from air-breathing predators." But seals are deepening their dives before the arrival of phytoplankton subsidies! Therefore, the data cannot support that statement. Should this be reframed as a hypothesis based on in situ production of phytoplankton? Or is that implausible because "local break-out occurs substantially later than seal dive shallowing"?

Thank you for pointing this out. We have removed the potential mechanism ("to take advantage of the new phytoplankton availability") and left the description of the pattern

intact. The sentence now reads: **“Foraging depth of Weddell seals halved and remained shallow for three weeks in mid-summer, suggesting that zooplankton and fishes may have shallowed (57), despite the predation risk from air-breathing predators (Figure 6)”**.

Minor comments

Line 208. Apparent typo: “proceeded” vs “preceeded”

Thank you for noticing our typo. We have changed “proceeded” to “progressed”.

Lines 237-239. There is a wealth of literature supporting the notion that bottom time (and, potentially, net energy gain) increases with the shallowing of dives. You might consider citing one of those papers here.

We appreciate this suggestion, and have added a citation to the end of the sentence (Zimmer, Ilka, et al. "Dive efficiency versus depth in foraging emperor penguins." *Aquatic Biology* 8.3 (2010): 269-277.)

Figs 2 and 3. The term “ice” is attached to each location name on the text embedded in the figure (And table in Fig3). This is confusing, as the locations are associated with the timing of both ice break-out and advection. Is the term “ice” really meant to be there?

Thank you for pointing out this redundancy. We have removed the term “ice” from all of Figures 2 and 3.

Fig 3 The map should include labels for Cape Royds, Erebus Bay, Ross Island

We have added labels for Cape Royds, Erebus Bay, and Ross Island to the map. Please note that based on feedback from Reviewer 1, this is now Figure 2 in the manuscript.

Fig 5. Caption should specify that each line in panels B and C represents an individual seal.

We have added the following sentence to the end of the Figure 5 legend: “each line in panels B and C represents an individual seal”.

References

1. Lima, S. L. Putting predators back into behavioral predator–prey interactions. *Trends Ecol. Evol.* 17, 70–75 (2002).
2. Frid, A., Burns, J., Baker, G. G. & Thorne, R. E. Predicting synergistic effects of resources and predators on foraging decisions by juvenile Steller sea lions. *Oecologia* 158, (2009).